# CLIP AS MULTI-TASK MULTI-KERNEL LEARNING

## ABSTRACT

Contrastive Language-Image Pretraining (CLIP) is a foundational model that learns a latent embedding space through an inner product-based objective. In this paper, we provide a theoretical interpretation of CLIP utilizing the Reproducing Kernel Hilbert Space (RKHS) framework. Specifically, we reformulate the problem of estimating the infinite-dimensional mapping $\phi$ with a neural network as selecting an unknown RKHS using multiple kernel learning. Such connection motivates us to propose to estimate the CLIP embedding via the multi-task multi-kernel (MTMK) method: we reformulate the different labels in the CLIP training data as the multiple training tasks, and reformulate learning the unknown CLIP embedding as choosing an optimal kernel from a family of Reproducing Kernel Hilbert Spaces, which is computationally more efficient. Utilizing the MTMK interpretation of CLIP, we also show an optimal statistical rate of the MTMK classifier under the scenario that both the number of covariates and the number of candidate kernels can increase with the sample size. Besides the synthetic simulations, we apply the proposed method to align the medical imaging data with the clinical codes in electronic health records and illustrate that our approach can learn the proper kernel space aligning the imaging embedding with the text embeddings with high accuracy.

## 1 INTRODUCTION

Contrastive Language-Image Pretraining (CLIP), introduced by Radford et al. [56], is a foundational model that connects natural language understanding and computer vision. CLIP has demonstrated excellent transferability across various tasks, including text-guided image generation with GANs [54; 13; 15], diffusion models [52; 57; 27; 59; 5] and image captioning [46; 76; 10]. Additionally, CLIP has shown its adaptability to various languages [73; 9] and 2D art forms [70]. Furthermore, the mechanisms behind CLIP extend to broader domains such as 3D object generation [24; 20; 45] and audio synthesis [18]. These aspects of CLIP have sparked the interest of the community, and researchers are now actively seeking a deeper understanding while expanding its boundaries [39; 71; 75; 77; 47].

At its core, CLIP operates by jointly optimizing a text encoder and an image encoder to learn a latent embedding space $\mathcal{H}$ by contrastive learning [8]. CLIP maximizes the cosine similarity of the $N$ real pairs in the diagonal and minimizes the cosine similarity for the $N^2 - N$ incorrect pairs. Intuitively, the contrastive objective of CLIP is to learn a similarity metric that aligns multi-modal inputs on $\mathcal{H}$, which is the inner product of the mapping $\phi$. This motivates us to consider $\mathcal{H}$ as an unknown Reproducing Kernel Hilbert Space (RKHS) where its kernel represents the measure of similarity. Formally, let $\boldsymbol{X} = (X_1, \ldots, X_k)^T$ be a $k$-dimensional random vector in $\mathcal{X}^k$ and $Y$ be a categorical random variable in $\mathcal{Y} = \{1, \ldots, T\}$. Given a sample pair $(\boldsymbol{x}_i, y_i)$ and $(\boldsymbol{x}_j, y_j)$, we can define the contrastive learning objective $C_{\mathcal{H}}$ of CLIP in the RKHS $\mathcal{H}$ as

$$C_{\mathcal{H}}(\boldsymbol{x}_i, \boldsymbol{x}_j) := \langle \phi(\boldsymbol{x}_i), \phi(\boldsymbol{x}_j) \rangle_{\mathcal{H}} (\mathbb{I}\{y_i = y_j\} - \mathbb{I}\{y_i \neq y_j\}),$$

where $\phi(\cdot)$ denotes the feature mapping and $\langle \cdot, \cdot \rangle_{\mathcal{H}}$ denotes the inner product in $\mathcal{H}$. Within this context, CLIP can be perceived as learning an unknown RKHS $\mathcal{H}$ by maximizing the contrastive objective $\max_{\mathcal{H} \in \mathbb{H}} C_{\mathcal{H}}^{\mathcal{D}}$, where $\mathcal{D} = \{(\boldsymbol{x}_i, y_i)\}_{i=1}^n$ denotes the training samples and $\mathbb{H}$ denotes the family of RKHSs. For instance, when the feature map $\phi$ is estimated by neural networks, its induces the unknown RKHS $\mathcal{H}$ via the neural tangent kernel (NTK) theory [22].

Although CLIP has an intuitive connection to RKHS, there are several gaps in our understanding of the mechanisms behind it. For instance, while there are general theoretical results regarding

contrastive learning [1] and kernel learning [3], CLIP lacks a specific theoretical framework that illustrates or explains its generalization capability. Although the community gained progress in devising data-efficient CLIP [39; 71], there is no formal quantification of its learning rate, leaving researchers with limited insights. In this paper, we address these gaps by presenting an RKHS approach to understanding CLIP's generalization properties. While we delayed the detailed setting and definition to Section 2, we present a brief preview here. We formulate the learning objective of CLIP in a multi-task binary setting. In each task $t = 1, \ldots, T$, we consider the following generalized version of the objective:

$$\max_{\mathcal{H} \in \mathbb{H}} \max_{\boldsymbol{\alpha}_t} \sum_{i=1}^{n} \ell\Big( \sum_{j=1}^{n} \alpha_{tj} \boldsymbol{C}_{\mathcal{H}}(\boldsymbol{x}_{ti}, \boldsymbol{x}_{tj}) \Big),$$

where $\boldsymbol{\alpha}_t = (\alpha_{t1}, \ldots, \alpha_{tn})^T \in \mathbb{R}^n$ is a weight vector, $\ell$ is a loss function and $\mathbb{H}$ is a specific family of RKHSs. Through the above loss, we reformulate the CLIP encoder estimating as identifying the optimal $\mathcal{H}$ from $\mathbb{H}$. With this formulation, we show that CLIP can be associated to a multi-task multi-kernel learning problem

$$\max_{\boldsymbol{\alpha}'_t} \sum_{t=1}^{T} \sum_{i=1}^{n} \ell\big( y_{ti} f_t(\boldsymbol{x}_{ti}) \big) - \text{pen}(f),$$

where $f_t \in \mathcal{H}$ is a classifier function for task $t$, $f$ is the set of classifiers of all tasks, and pen is a penalty function regularizing the nonparametric classifiers. We further prove the statistical rate of the classifier estimator when $\ell$ is selected as the logistic loss. We conclude our main contributions as the following:

- We reformulate the notion of CLIP by a solid RKHS framework. CLIP and its variants are assumed to learn an unknown encoder map by maximizing a contrastive objective. In this setting, we show that the unknown map corresponds to a feature map in an unknown RKHS. We further propose to learn the unknown RKHS by multi-kernel learning and formalize CLIP as multi-task multi-kernel learning.

- Our methodological contribution is to reduce the problem of estimating the optimal mapping $\phi$ to selecting an optimal RKHS by multiple kernel learning. Moreover, although we focus on the analysis of CLIP, our method can be extended to any representation learning that involves estimating $\phi$ via the optimization of an inner product objective.

- Under this formalization, we prove that our multi-task multi-kernel learner has an optimal convergence rate compared to classic linear models and nonparametric models, which involves the analysis of a generalized version of the empirical V-statistics process [19]. We further demonstrate that when the optimal RKHS is a sparse combination of candidate kernels, our model involves an implicit model selection, where the number of prospective candidate models can be much greater than the sample size.

In summary, this paper contributes to understanding the robust transferability of CLIP while providing insights into its theoretical foundation via the RKHS framework. It paves the way for further research and innovation in the field of statistics and artificial intelligence.

## 1.1 RELATED LITERATURE

The existing approaches to theoretical understanding of CLIP's mechanism can be categorized as two major types. One line of work analyzes the properties of the representation learned by CLIP like modality gap [40]. Another line of work attempts to reduce CLIP to some well-studied problems for mapping $\phi$ to the finite dimensional spaces [68; 49], while we consider the infinite dimensional mapping.

Our work also concerns multiple kernel learning (MKL), where numerous papers have studied various formulations of the linear combination of kernels [36; 69; 29; 30; 72; 3; 2; 61; 62; 66]. A more relevant line of research exploits the group structure of kernels in MKL [66; 23; 64; 67; 31; 25; 79; 48]. However, though there are some analyses of the excess risk [36; 31], few MKL research concentrates on the statistical rate of convergence. Another line of research conducts extensive theoretical analyses of MKL in the larger context of sparse additive model [34; 44; 35; 65]. But there is no analysis conducted on classification problems in this line, not to mention multiclass classification. Furthermore, no theoretical analysis of MKL has been conducted in the multi-task setting.

In addition, kernel methods have been broadly applied to multiple task learning (MTL) [16; 78; 80; 14; 11; 12; 26]. We refer to Zhang & Yang [81] for a recent survey. Another line of research investigates sparse linear models in MTL [41; 53; 32; 33; 50; 51]. Existing theoretical analyses are conducted on linear models, while no theoretical guarantees are made in nonparametric models. Moreover, there is no concrete formulation of how to perform multiple kernel selection in MTL. Our approach, regarding logistic regression, bridges both gaps by using a group Lasso penalty for joint kernel selection in a multi-task setting and obtains sharp convergence rates comparable to the linear model in Lounici et al. [41] and single-task MKL in Suzuki & Sugiyama [65].

## 1.2 NOTATION

Let $\mathbb{I}\{\cdot\}$ be the indicator function. For two positive sequences $\{x_n\}_{n=1}^{\infty}$ and $\{y_n\}_{n=1}^{\infty}$, we say $x_n = O(y_n)$ if $x_n \leq Cy_n$ holds for any $n$ with some large enough $C > 0$. We use the notation $x_n \asymp y_n$ if $x_n = O(y_n)$ and $y_n = O(x_n)$. And we say $x_n = o(y_n)$ if $x_n/y_n \to 0$ as $n \to \infty$. For a sequence of random variables $\{X_n\}_{n=1}^{\infty}$ and a corresponding set of constants $a_n$, denote $X_n = O_{\mathbb{P}}(a_n)$ if for all $\epsilon > 0$, there exists a finite $M > 0$ and $N > 0$ such that $\mathbb{P}(|X_n/a_n| > M) < \epsilon$ for all $n > N$. And for a scalar $a$, we say $X_n \leq a + o_{\mathbb{P}}(1)$ if for all $\epsilon > 0$, $\lim_{n\to\infty}\mathbb{P}(X_n - a > \epsilon) = 0$. For a vector $\boldsymbol{a} \in \mathbb{R}^k$, we denote the $q$ norm as $\|\boldsymbol{a}\|_q = \sum_i(|a_i|^q)^{1/q}$. For the function $f \in L^2(\Pi)$ where $\Pi$ is the probability metric, we define the $L^2$ norm $\|f\|_{L_2(\Pi)} = [\int f(\boldsymbol{x})^2\mathrm{d}\Pi(\boldsymbol{x})]^{1/2}$ and the supremum norm $\|f\|_{\infty} = \sup_{\boldsymbol{x}\in\mathbb{R}^k}|f(\boldsymbol{x})|$. Throughout this paper, $c, C, C_1, C_{\beta}, \dots$ are used as generic constants whose values may vary across different places.

## 2 CLIP AS MULTI-TASK LEARNING IN RKHS

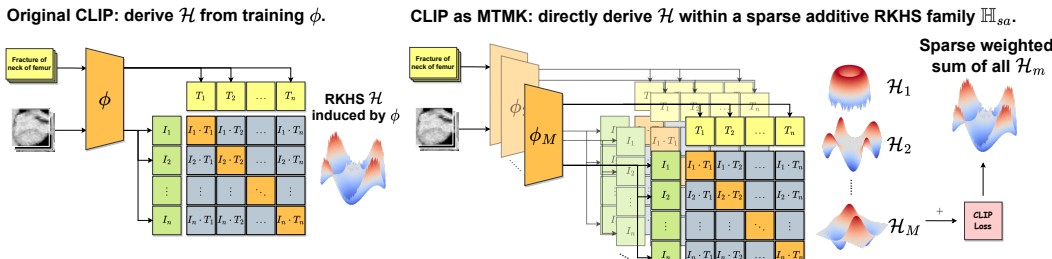

Figure 1: Original CLIP derives its RKHS $\mathcal{H}$ through training $\phi$. In our formulation, we directly obtain $\mathcal{H}$ as a sparse weighted combination of all candidate RKHSs.

We begin with a brief overview of the general connections between neural networks and RKHS. Jacot et al. [22] proves that the training process of a certain family of neural networks is equivalent to kernel regression with the NTK. Recent research has further explored the close connection between neural networks and standard kernels [7]. We now shift our focus to CLIP. CLIP learns a latent embedding space for text and images through contrastive loss, such that text-image pairs in the diagonal are maximized while all other pairs are minimized. Specifically, CLIP learns a mapping $\phi$ for both text and images in an unknown RKHS $\mathcal{H}$ by maximizing the inner product within the same class and minimizing it between samples from different classes. Let $\boldsymbol{X} = (X_1, \dots, X_k)^T$ be a $k$-dimension random vector in $\mathcal{X}^k$ and $Y$ be a categorical random variable in $\mathcal{Y} = \{1, \dots, T\}$. Given a sample pair $(\boldsymbol{x}_i, y_i)$ and $(\boldsymbol{x}_j, y_j)$, we can define the contrastive learning objective in the RKHS $\mathcal{H}$ as

$$\boldsymbol{C}_{\mathcal{H}}(\boldsymbol{x}_i, \boldsymbol{x}_j) := \langle\phi(\boldsymbol{x}_i), \phi(\boldsymbol{x}_j)\rangle_{\mathcal{H}}(\mathbb{I}\{y_i = y_j\} - \mathbb{I}\{y_i \neq y_j\}), \qquad (2.1)$$

where $\phi(\cdot)$ denotes the feature mapping and $\langle\cdot,\cdot\rangle_{\mathcal{H}}$ denotes the inner product in $\mathcal{H}$. CLIP attempts to find the optimal RKHS $\mathcal{H}$ by maximizing the objective $\max_{\mathcal{H}\in\mathbb{H}} \boldsymbol{C}_{\mathcal{H}}^{\mathcal{D}}$, where $\mathcal{D} = \{(\boldsymbol{x}_i, y_i)\}_{i=1}^n$ is the training set and $\mathbb{H}$ is a family of RKHSs. Note that we assume the pseudo labels of sample pairs used by CLIP can be treated as labels in our analysis. A prevalent approach to RKHS learning involves maximizing $\sum_{i=1}^n\sum_{j=1}^n \boldsymbol{C}_{\mathcal{H}}(\boldsymbol{x}_i, \boldsymbol{x}_j)$, which is known as the kernel alignment method [17; 21; 55]. In practice, CLIP optimizes a symmetric cross entropy loss for both text and images, where row-wise and column-wise maximization is carried out separately. In addition, CLIP involves

a temperature parameter to weight the logits. Therefore, we consider the following CLIP loss

$$\max_{\mathcal{H}\in\mathbb{H}} \max_{\boldsymbol{\alpha}} \sum_{i=1}^{n} \ell\Big( \sum_{j=1}^{n} \alpha_j \boldsymbol{C}_{\mathcal{H}}(\boldsymbol{x}_i, \boldsymbol{x}_j) \Big), \tag{2.2}$$

where $\ell$ represents a general loss function, such as softmax and logistic loss and $\boldsymbol{\alpha} = (\alpha_1, \ldots, \alpha_n)^T$ is the weight vector generalizing the temperature parameter. The inclusion of $\ell$ provides a more comprehensive understanding, as it can be reduced to the kernel alignment approach when $\ell$ is set as the identity function. We search for the optimal weighting for the objective function by maximization over $\boldsymbol{\alpha}$ and estimate the desirable RKHS $\mathcal{H}$ within a family of RKHSs $\mathbb{H}$.

Unfortunately, it is not attainable to analyze an infinitely large family of RKHSs. A commonly used structural assumption on $\mathbb{H}$ is that it is the span of given candidate spaces

$$\mathbb{H}_a = \Big\{ \mathcal{H} : \langle \phi(\boldsymbol{x}_i), \phi(\boldsymbol{x}_j) \rangle_{\mathcal{H}} = \sum_{m=1}^{M} \omega_m \langle \phi_m(\boldsymbol{x}_i), \phi_m(\boldsymbol{x}_j) \rangle_{\mathcal{H}_m}, \omega_m \geq 0 \Big\},$$

where $\mathcal{H}_m$ is an RKHS with a prespecified kernel such as Gaussian kernel, polynomial kernel, and NTK. Additionally, $\mathcal{H}_m$ is also a subspace of the $L_2(\Pi)$ space. Given that neural networks can induce high non-linearity, the number of candidate RKHSs $M$ should be sufficiently large to construct a search space comparable to that of deep models. However, this is not feasible in a dense setting due to the curse of dimensionality. Hence, we focus on the analysis of the sparse additive family

$$\mathbb{H}_{sa} = \Big\{ \mathcal{H} : \langle \phi(\boldsymbol{x}_i), \phi(\boldsymbol{x}_j) \rangle_{\mathcal{H}} = \sum_{m\in I} \omega_m \langle \phi_m(\boldsymbol{x}_i), \phi_m(\boldsymbol{x}_j) \rangle_{\mathcal{H}_m}, \omega_m > 0 \Big\}, \tag{2.3}$$

where $I \subset [M]$ is the index set and $d := |I| \ll M$. Within this specific family of RKHSs, we are able to analyze the given problem (2.2). For the contrastive objective defined in (2.1), its form can be difficult to interpret in multiclass classification. To gain more insight, we first establish the optimization problem in the binary case. For $\mathcal{Y} = \{+1, -1\}$, we can rewrite the problem (2.2) as

$$\max_{\mathcal{H}\in\mathbb{H}_{sa}} \max_{\boldsymbol{\alpha}} \sum_{i=1}^{n} \ell\Big( \sum_{j=1}^{n} \alpha_j y_i y_j \langle \phi(\boldsymbol{x}_i), \phi(\boldsymbol{x}_j) \rangle_{\mathcal{H}} \Big). \tag{2.4}$$

We then reparameterize the weights in (2.4) as $\widetilde{\boldsymbol{\alpha}} := \boldsymbol{y} \odot \boldsymbol{\alpha}$ where $\odot$ represents the element-wise multiplication, which leads to

$$\max_{\mathcal{H}\in\mathbb{H}_{sa}} \max_{\widetilde{\boldsymbol{\alpha}}} \sum_{i=1}^{n} \ell\Big( y_i \sum_{j=1}^{n} \sum_{m=1}^{M} \alpha'_{mj} \langle \phi_m(\boldsymbol{x}_i), \phi_m(\boldsymbol{x}_j) \rangle_{\mathcal{H}_m} \Big)$$
$$\text{s.t. } \alpha'_{mj} = \omega_m \widetilde{\alpha}_j \tag{2.5}$$

where $\langle \phi(\boldsymbol{x}_i), \phi(\boldsymbol{x}_j) \rangle_{\mathcal{H}} = \sum_{m=1}^{M} \omega_m \langle \phi_m(\boldsymbol{x}_i), \phi_m(\boldsymbol{x}_j) \rangle_{\mathcal{H}_m}$ by (2.3). We further relax the constraint (2.5) to have

$$\max_{\boldsymbol{\alpha}'} \sum_{i=1}^{n} \ell\Big( y_i \sum_{j=1}^{n} \sum_{m=1}^{M} \alpha'_{mj} \langle \phi_m(\boldsymbol{x}_i), \phi_m(\boldsymbol{x}_j) \rangle_{\mathcal{H}_m} \Big). \tag{2.6}$$

In Section 4, we present a heuristic method to recover the the additive structure of $\mathcal{H}$. Now, it seems that we can settle with studying (2.6) for binary classification, but this deviates from the original CLIP which performs an $T$ class classification within each batch. To address this issue, we propose to transform the $T$ class classification problem into a multi-task binary classification via the one-vs-rest or one-vs-one strategy, which is a simple and effective solution. As such, we establish the problem as follows:

$$\max_{\boldsymbol{\alpha}'_t} \sum_{t=1}^{T} \sum_{i=1}^{n} \ell\Big( y_{ti} \sum_{j=1}^{n} \sum_{m=1}^{M} \alpha'_{mtj} \langle \phi_m(\boldsymbol{x}_{ti}), \phi_m(\boldsymbol{x}_{tj}) \rangle_{\mathcal{H}_m} \Big), \tag{2.7}$$

where $t = 1, \ldots, T$ denotes each task. Although we introduce multi-task learning to serve as a potential solution to the challenges associated with multiclass classification, it embodies broader applicability to scenarios where tasks are related but not necessarily parallel.

However, CLIP utilizes only one mapping for each class, which means the same set of kernels needs to be shared between each task. This motivates us to introduce a penalty term pen with regards to $\boldsymbol{\alpha}'_t$. Hence, according to the representer theorem [28], (2.7) is equivalent to

$$\max_{\boldsymbol{\alpha}'_t} \sum_{t=1}^{T} \sum_{i=1}^{n} \ell(y_{ti} f_t(\boldsymbol{x}_{ti})) - \text{pen}(f),$$

where $f_t(\cdot) = \sum_{j=1}^{n} \sum_{m=1}^{M} \alpha'_{mtj} \langle \phi_m(\cdot), \phi_m(\boldsymbol{x}_{tj}) \rangle_{\mathcal{H}_m}$ and $f$ is the set of classifiers of all tasks. We will discuss the specific form of the penalty in the next section. Without the loss of generality, we set up the problem with the logistic loss as $\ell$, which gives the following objective

$$\min_{\boldsymbol{\alpha}'_t} \sum_{t=1}^{T} \sum_{i=1}^{n} \log\left(1 + \exp\left(-y_{ti} f_t(\boldsymbol{x}_{ti})\right)\right) + \text{pen}(f).$$

In summary, we reduce CLIP to a multi-task multi-kernel classification problem. In this formulation, we can study the estimation rate of a multi-task multi-kernel logistic regression.

## 3 MULTI-TASK MULTI-KERNEL LOGISTIC REGRESSION

In this section, we investigate the problem of estimating the multi-task multi-kernel logistic regression under structured sparsity assumptions on the underlying kernels. The aim is to jointly estimate truly active kernel functions across all tasks with an appropriate selection of the penalty term. We denote $t = 1, \ldots, T$ as independent tasks, $m = 1, \ldots, M$ as candidate kernels and the true classifier as $f^*$. $f^*$ belongs to an RKHS $\mathcal{H}$ in the sparse additive family $\mathbb{H}_{sa}$. We also define $I_0$ as the subset of kernels such that $f^*_{mt} \neq 0$ for $m \in I_0$ and for all $t = 1, \ldots, T$. Given a $k$-dimension random vector $\boldsymbol{X}_t = (X_1, \ldots, X_k)^T$ in $\mathcal{X}^k$ and a binary response variable $Y_t \in \{-1, +1\}$, we assume

$$\log \frac{\mathbb{P}(Y_t = 1 \mid \boldsymbol{X}_t)}{\mathbb{P}(Y_t = -1 \mid \boldsymbol{X}_t)} = \sum_{m \in I_0} f^*_{mt}(\boldsymbol{X}_t). \tag{3.1}$$

The unknown function $f^*_{mt}$ is usually estimated by the maximum likelihood estimator (MLE), that is, we minimize the following negative log-likelihood function with respect to $f_{mt}$ across all tasks:

$$L(f) = \sum_{t=1}^{T} \sum_{i=1}^{n} \log\left(1 + \exp\left(-y_{ti} f_t(\boldsymbol{x}_{ti})\right)\right),$$

where $f_t(\boldsymbol{x}_{ti}) = \sum_{m=1}^{M} f_{mt}(\boldsymbol{x}_{ti})$.

To exploit sparsity in our setting, we consider the $\ell_1$ penalty and the elastic-net type penalty. In particular, for the function $f_{mt}$ in the RKHS $\mathcal{H}_m$, we employ $\|f_{mt}\|_n$ and $\|f_{mt}\|_{\mathcal{H}_m}$ as $\ell_1$ regularizers and $\|f_{mt}\|_{\mathcal{H}_m}^2$ as $\ell_2$ regularizers. Here, we define $\|f\|_n = [\frac{1}{n} \sum_{i=1}^{n} f(\boldsymbol{x}_i)^2]^{1/2}$ and $\|f_{mt}\|_{\mathcal{H}_m}$ is the RKHS norm of $f_{mt}$ in $\mathcal{H}_m$. With these notations, we can set the penalty term as follows:

$$\text{pen}(f) = \lambda_1 \sum_{t=1}^{T} \sum_{m=1}^{M} \|f_{mt}\|_n + \lambda_2 \sum_{t=1}^{T} \sum_{m=1}^{M} \|f_{mt}\|_{\mathcal{H}_m} + \lambda_3 \sum_{t=1}^{T} \sum_{m=1}^{M} \|f_{mt}\|_{\mathcal{H}_m}^2.$$

The above model utilizes a common combination of $\ell_1$ and $\ell_2$ penalties, which is analyzed in Koltchinskii & Yuan [34]; Raskutti et al. [58]; Suzuki & Sugiyama [65], though they study the problem of linear regression or support vector machine rather than logistic regression. However, the above model only considers a plain summation of all tasks, which gives the same theoretical rate as in single-task MKL. Thus, it may be suboptimal in a multi-task setting for ignoring the inner connection between tasks. Moreover, to be consistent with CLIP, the RKHS $\mathcal{H}$ should share the same set of kernels across each task. Therefore, we introduce our new penalized multi-task multi-kernel estimator. The loss function is given as

$$L(f) = \sum_{t=1}^{T} \sum_{i=1}^{n} \log\left(1 + \exp\left(-y_{ti} f_t(\boldsymbol{x}_{ti})\right)\right) + \lambda_1 \sum_{m=1}^{M} \sqrt{\sum_{t=1}^{T} \|f_{mt}\|_n^2}$$

$$+ \lambda_2 \sum_{m=1}^{M} \sqrt{\sum_{t=1}^{T} \|f_{mt}\|_{\mathcal{H}_m}^2} + \lambda_3 \sum_{m=1}^{M} \sum_{t=1}^{T} \|f_{mt}\|_{\mathcal{H}_m}^2. \tag{3.2}$$

Here, our $\ell_1$ regularizer is a special case of the group Lasso penalty. This mixed $2, 1$ regularizer takes advantage of the kernel-wise structure and encourages consistent sparsity patterns across all tasks.

Under this penalty, we have either $\widehat{f}_{mt} = 0$ for all $t$ or $\widehat{f}_{mt} \neq 0$ for all $t$. We obtain our proposed estimator as

$$\widehat{f} = \underset{\substack{f_{mt} \in \mathcal{H}_m(C) \\ m \in [M], t \in [T]}}{\operatorname{argmin}} L(f), \qquad (3.3)$$

where $\mathcal{H}_m(C) := \{f_{mt} | \sum_{m=1}^M \|f_{mt}\|_{\mathcal{H}_m} \leq C\}$ for some $C$ sufficiently large.

## 4  METHOD

In this section, we discuss the optimization of the objective function (3.2). By the representer theorem [28] and the kernel trick, we can express the solution $\widehat{f}_{mt}$ as a linear combination of kernels: $\widehat{f}_{mt}(\boldsymbol{x}) = \sum_{i=1}^n \alpha_{mti} k_m(\boldsymbol{x}_{ti}, \boldsymbol{x})$. Thus, we can perceive each kernel as a block and consider the block coordinate gradient descent (BCGD) algorithm [43]. The key idea of BCGD is to combine a second-order approximation of the log-likelihood with line search and update $\boldsymbol{\alpha}_{mt}$ in a block-wise manner. A detailed discussion of BCGD can be found in Section B of supplementary materials.

Due to convex relaxation in (2.6), we are not able to directly derive the inner product $\langle \phi, \phi \rangle_{\mathcal{H}}$, but we provide a heuristic approach to dealing with this issue. Given that the sparse additive family has the following structure: $\langle \phi(\boldsymbol{x}_i), \phi(\boldsymbol{x}_j) \rangle_{\mathcal{H}} = \sum_{m \in I} \omega_m \langle \phi_m(\boldsymbol{x}_i), \phi_m(\boldsymbol{x}_j) \rangle_{\mathcal{H}_m}$, we only need to recover $\boldsymbol{\omega}$ using the property of kernels within the sparse index set $I$ defined in (2.3). Recall that (2.6) gives $\alpha_{mti} = \omega_{mt} \alpha_{ti} y_{ti}$, where $\alpha_{mti}$ is the coefficient attained by solving (3.2) with BCGD. Thus, we can perform matrix factorization on $\boldsymbol{\alpha}_t / \boldsymbol{y}_t$ (where "/" represents the element-wise division) and extract the resulting vector as $\boldsymbol{\omega}$. For example, we consider the non-negative matrix factorization of $\boldsymbol{\alpha}_t / \boldsymbol{y}_t$, which minimizes $\|\boldsymbol{\alpha}_t / \boldsymbol{y}_t - \boldsymbol{W} \boldsymbol{H}\|_F^2$ and $\| \cdot \|_F$ denotes the Frobenius norm, $\boldsymbol{W} \in \mathbb{R}^{M \times p}$ and $\boldsymbol{H} \in \mathbb{R}^{p \times n}$ are approximating matrices. We can specify $p = 1$ and use the resulting $\boldsymbol{W}$ to approximate the real weight vector $\boldsymbol{\omega}$.

## 5  ASSUMPTIONS

In this section, we describe several assumptions used in our theoretical analysis.

**Assumption 5.1** (Bounded Kernel). For each $m = 1, \ldots, M$, $\mathcal{H}_m$ is separable and $\sup_{X \in \mathcal{X}} |k_m(X, X)| \leq 1$.

The assumption above indicates that we only consider RKHSs of bounded kernels, which gives the relation $\|f_{mt}\|_\infty \leq \|f_{mt}\|_{\mathcal{H}_m}$ [63].

**Assumption 5.2** (Union Bound Assumption). There exists a constant $C_\beta$ such that for all $t = 1, \ldots, T$, we have $\sum_{m=1}^M \|f_{mt}^*\|_{\mathcal{H}_m} \leq C_\beta$.

This assumption simply provides a bound of the second derivative of the log-likelihood and can be relaxed to more general assumptions.

Next, we depict the complexity of RKHSs with the following assumption.

**Assumption 5.3** (Spectral Assumption). There exists $0 < s < 1$ and and some constant $c$, such that for any $j \geq 1$ and any $m = 1, \ldots, M$, we have $\mu_{j,m} \leq c j^{-\frac{1}{s}}$, where $\mu_{j,m}$ is the spectrum of the kernel $k_m$.

The Spectral Assumption ensures the average entropy number assumption [63], and it provides a more intuitive insight into the smoothness of the RKHSs by specifying the polynomial eigenvalue decay rate [42]. We also introduce some measures to characterize the dependence between the spaces $\mathcal{H}_m$, and impose the Incoherence Assumption [34; 44; 65]. See Section A of supplementary materials for details. In addition, we introduce the following technical assumption which gives an upper bound for $\|f_{mt}\|_\infty$ in terms of $\|f_{mt}\|_{L_2(\Pi)}$ and $\|f_{mt}\|_{\mathcal{H}_m}$.

**Assumption 5.4** (Sup-norm Assumption). Along with the Spectral Assumption, for all $f_{mt} \in \mathcal{H}_m, m = 1, \ldots, M$ and $t = 1, \ldots, T$, there exists a constant $C_1$ such that $\|f_{mt}\|_\infty \leq C_1 \|f_{mt}\|_{L_2(\Pi)}^{1-s} \|f_{mt}\|_{\mathcal{H}_m}^s$, where $s$ is the exponent defined in the Spectral Assumption.

It seems that this assumption may be a bit strong for specifying the same $s$ defined in the Spectral Assumption. However, this condition is satisfied if the RKHS is either a Sobolev space or continuously embedded in a Sobolev space [65]. Therefore, kernels in practical use usually satisfy this condition which gives a sharper bound on $\|f_{mt}\|_\infty$.

## 6 MAIN RESULT

In this section, we present the theoretical result of our proposed estimator in (3.3) and give some remarks. The following theorem describes the convergence rate under assumptions in Section 5.

**Theorem 6.1.** Suppose all assumptions are satisfied and there exists some constant $c > 0$ such that $\|f_{mt}^*\|_{\mathcal{H}_m} \geq c$ for all $m \in I_0$ and $t = 1, \ldots, T$. Set $\lambda_1 = C\sqrt{\frac{T + \sqrt{T}\log M}{n^{1/(1+s)}}}$, $\lambda_2 = C\sqrt{\frac{T + \sqrt{T}\log M}{n^{2/(1+s)}}}$, $\lambda_3 = C\frac{1}{n^{1/(1+s)}}$ or $\lambda_3 = 0$ for some $C$ sufficiently large. Under the condition that $\frac{d\log M}{\sqrt{n}} + \frac{d}{n^{(1-s)/2(1+s)}} = o(1)$, the estimator $\widehat{f}_t$ satisfies

$$\frac{1}{T}\sum_{t=1}^{T}\|\widehat{f}_t - f_t^*\|_{L_2(\Pi)}^2 \leq O_{\mathbb{P}}\left(\frac{d}{n^{1/(1+s)}}\left(1 + \frac{\log M}{\sqrt{T}}\right)\right). \tag{6.1}$$

The estimation error comes from two sources. The empirical process of $f_{mt}$ contributes $O(d/n^{1/(1+s)})$ in (6.1) while the second term $O(\log M/\sqrt{T})$ is introduced by using $T$ tasks to jointly estimate the active kernels from $M$ candidates. At the core of our approach, the convergence rate depends on a sharp upper bound for $|\frac{1}{n}\sum_{t=1}^{T}\sum_{i=1}^{n}\epsilon_{ti}f_{mt}(\boldsymbol{x}_{ti})|$, where $\epsilon_{ti}$ is the gradient of the log-likelihood. Although $t$ and $i$ are both independent, the above statistic cannot be simply interpreted as a double sum of $nT$ independent points, rendering it a generalized empirical V-statistics process [19]. It is generalized in the sense that $\epsilon_{ti}$ and $f_{mt}(\boldsymbol{x}_{ti})$ do not conform to an identical distribution. Detailed proof can be found in Section E of supplementary materials.

We obtain the optimal theoretical rate for estimating the nonparametric model in a multi-task setting. If $s \to 0$ for all RKHSs, the Spectral Assumption restricts the kernels to be finite rank ($\mu_{j,m} \to 0$ for $j \geq 2$ and $\mu_{1,m} = c$ for some $c \in \mathbb{R}$) and our model essentially becomes a linear model. Thus, our model can be compared with the multi-task linear model considered in Lounici et al. [41]. They give a non-asymptotic rate at $\frac{d}{n}(1 + \frac{\log M}{\sqrt{T}})$ for their group Lasso type estimator, where the structured sparsity assumption is imposed on features. Moreover, the nonparametric estimator is known to have an optimal scale of $n^{1/(1+s)}$ [58]. We derive the identical convergence rate with regard to $n$ under the same assumption of polynomial eigenvalue decay.

Now we shed some light on the relations between these parameters. It is natural in multi-task learning to assume $T \geq n$. Thus, the bounds in Theorem 6.1 become independent of $M$ given $\log M \leq \sqrt{T}$. This indicates that we can construct an optimal RKHS with an unknown kernel from a huge RKHS candidate pool given that truly active kernels are sparse. For example, we can select $M = \exp(n^\gamma)$ for $\gamma < 1/2$, provided that $T > n^{2\gamma}$. This ability admits promising expressiveness in finding a highly non-linear map for data points, which is a desired property common in deep learning models. With this theoretical result, we demonstrate that when the optimal RKHS is a sparse combination of candidates, our model involves an implicit model selection process with regard to kernels, where the number of prospective candidate models can be much greater than the sample size.

## 7 NUMERICAL RESULTS

### 7.1 SYNTHETIC DATA

In this section, we present some simulation results to demonstrate the performance of our MTMK models in two aspects. The first objective is to show how our proposed model suits a structured sparsity setting better than other models. The second objective is to empirically demonstrate the theoretical properties shown in Theorem 6.1. We consider 7 models for comparison: MTMK mix, MTMK L1, MTMK L2, STMK, MTSK, STSK and ORACLE, where M denotes multiple and S denotes single. MTMK mix and MTMK L1 are our proposed models with the elastic-net penalty and $\ell_1$ penalty, while MTMK L2 is MK ridge logistic regression. In STMK, the sparsity pattern does not share between tasks and MTSK is a ridge logistic regression with the Gaussian kernel $K(x, y) = \exp(\|x - y\|^2)$. STSK is theoretically the same as MTSK due to convexity but a minor numerical difference might be observed. Only true kernels are given as input in ORACLE, which indicates the best possible performance that can be achieved by the models listed above.

Data are drawn from a uniform distribution $X \sim U(-1, 1)$. Labels $\{+1, -1\}$ are generated based on the probability distribution $\mathbb{P}(Y = 1 \mid X) = 1/(\exp(-f(X)) + 1)$, where $f$ is the true function.

We use a sample size of 1000 and a task number of 6. For simplicity, $\lambda_1$ is omitted in the simulation experiment and we only consider $\lambda_2$ and $\lambda_3$. We perform grid search to tune hyperparameters based on likelihood on validation set. We manually construct 22 candidate kernels including Gaussian, sigmoid and polynomial kernels. In this experiment, we investigate three scenarios to evaluate the performance of our multi-task model, where signals of sparsity patterns varies between task sets. Detailed configuration is summarized in Section C of supplementary materials.

In each task set, we repeat the experiment 10 times with different random seeds and compute the standard deviation. Results are reported in Table 1, where we highlight the best-performing results in **bold** and the second-best results in underline. Our proposed model outperforms all other models in every task set, demonstrating results closest to those of ORACLE, the gold-standard model.

Table 1: Average mean squared error and standard deviation of 10 different random seeds.

| Model | Task Set 1 | Task Set 2 | Task Set 3 |
|---|---|---|---|
| ORACLE | $0.0100 \pm 0.0045$ | $0.0126 \pm 0.0052$ | $0.0217 \pm 0.0124$ |
| MTMK mix | $0.0184 \pm 0.0097$ | $0.0290 \pm 0.0105$ | $0.0448 \pm 0.0110$ |
| MTMK L1 | **$0.0184 \pm 0.0101$** | **$0.0260 \pm 0.0111$** | **$0.0347 \pm 0.0114$** |
| MTMK L2 | $0.0226 \pm 0.0103$ | $0.0291 \pm 0.0089$ | $0.0737 \pm 0.0138$ |
| MTSK | $0.1327 \pm 0.0109$ | $0.1309 \pm 0.0097$ | $0.3048 \pm 0.0203$ |
| STMK | $0.0248 \pm 0.0106$ | $0.0324 \pm 0.0108$ | $0.0476 \pm 0.0126$ |
| STSK | $0.1326 \pm 0.0107$ | $0.1309 \pm 0.0096$ | $0.3046 \pm 0.0203$ |

Next, we empirically investigate whether our proposed models possess the desirable properties outlined in our theoretical analysis. Specifically, we employ the identical settings as in task set 3 and only modify the parameter of interest. Results are summarized in Figure 2. Our models demonstrate enhanced approximation abilities as the candidate kernel size $M$ becomes smaller, task size $T$ and sample size $n$ become larger separately.

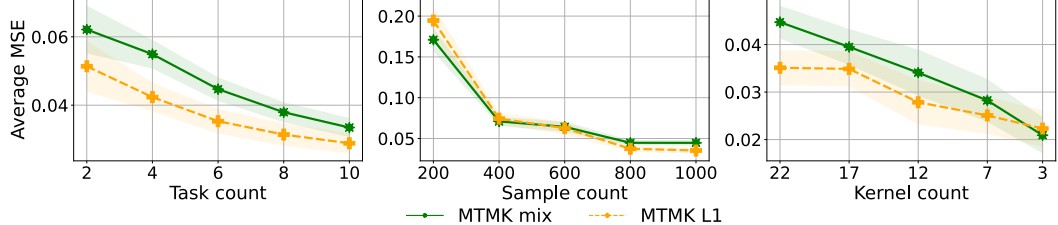

Figure 2: Effect of varying parameters on model approximation ability. Average mean squared error decreases as the candidate kernel size $M$ becomes smaller, task size $T$ becomes larger, and sample size $n$ becomes larger separately.

## 7.2 REAL DATA

In this section, we present an evaluation of our models on two widely-used real datasets, namely MNIST [37] and MedMNIST [74]. We utilize the one-vs-rest scheme to tackle the multi-class classification problem as in Li et al. [38]. Our analysis compares the models presented in section 7.1 and traditional machine learning models like SVM and logistic regression. For simplicity, candidate kernels are selected as Gaussian kernels and polynomial kernels. To comprehensively compare the performance of each model, we utilize the AUROC (Area Under the Receiver Operating Characteristic) metric. The result of MNIST is contained in Section D of supplementary materials.

We use the OrganCMNIST dataset, part of the MedMNIST collection, to evaluate our models. It consists of grayscale images of abdominal CT scans that depict 11 different organs, including the bladder, femurs, and nine other organs. We generate 200 samples for each task with balanced labels and remove repetitive organs, resulting in a total of 6 classification tasks of distinct organs.

Meanwhile, we also include phecodes of diseases related to the organs depicted in the images to showcase the multimodality of our proposed model. We obtain the word embeddings of these phecodes from the MIKGI dataset [82], which provides a 200-dimensional vector for each phecode. To incorporate these embeddings into our dataset, we simply add zero padding to convert them into

the same dimension as images. For each task, we include 50 related embeddings as positive samples and 50 unrelated embeddings as negative samples. We report AUROC results of images, texts, and all samples separately in Table 2. Our proposed models generally perform better than other benchmark models.

Table 2: MedMNIST AUROC with standard deviation over 5-fold cross-validation.

| Model | Average | Image | Text |
|---|---|---|---|
| MTMK mix | $0.9074 \pm 0.0156$ | $0.9310 \pm 0.0175$ | $0.8973 \pm 0.0310$ |
| MTMK L1 | $\mathbf{0.9102 \pm 0.0129}$ | $0.9357 \pm 0.0131$ | $\mathbf{0.9003 \pm 0.0278}$ |
| MTMK L2 | $0.9050 \pm 0.0172$ | $0.9278 \pm 0.0202$ | $0.8973 \pm 0.0306$ |
| MTSK | $0.8870 \pm 0.0121$ | $0.9104 \pm 0.0135$ | $0.8630 \pm 0.0261$ |
| STMK | $0.9053 \pm 0.0142$ | $\mathbf{0.9366 \pm 0.0128}$ | $0.8920 \pm 0.0339$ |
| SVM | $0.8267 \pm 0.0179$ | $0.9188 \pm 0.0235$ | $0.8380 \pm 0.0379$ |
| LR | $0.8879 \pm 0.0110$ | $0.8962 \pm 0.0112$ | $0.8970 \pm 0.0307$ |

We utilize Grad-CAM [60] to provide visual explanations for our model. Specifically, we select the MTMK L1 model trained on one fold of the MedMNIST dataset and randomly draw 10 positive image samples from each task for visualization. Each sample is presented with the original grayscale image and the gradient map overlaid on the image. In Figure 3(a), a clear heatmap pattern can be observed within one task, indicating that our model consistently extracts similar features for each organ. For example, MTMK L1 mainly focuses on the bottom left part of the lung classification task, capturing this organ's unique edge features. Furthermore, we demonstrate that our approach can learn the proper kernel space aligning the imaging embedding with the text embeddings with high accuracy. We randomly sample 3 phecodes that contain "heart" in the heart classification task. We compute the inner products with the method described in Section 4 between images and the phecodes, and present the top ten images with the largest inner products in Figure 3(b), where a green check represents a correct pairing and a red cross otherwise. Apparently, our model identifies the correct pairing most of the time.

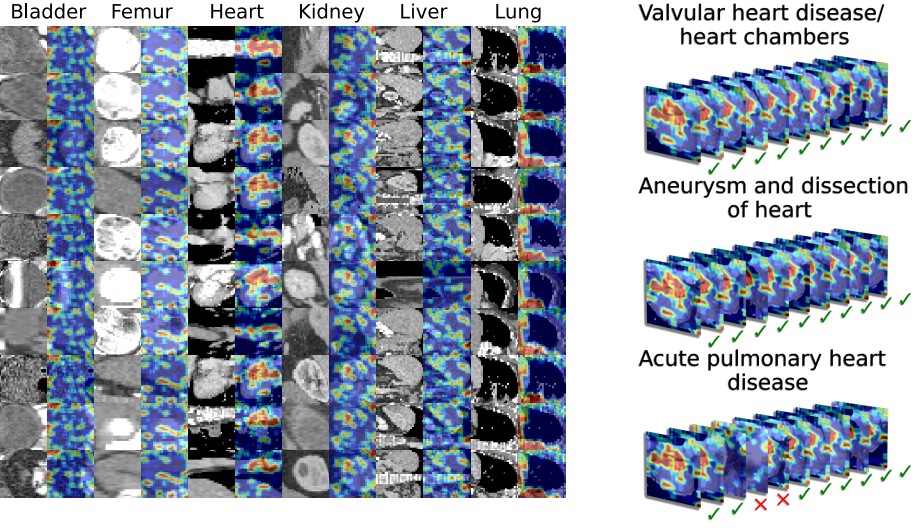

(a) Grad-CAM visualization.        (b) Inner product visualization.

Figure 3: Visualization for MedMNIST. (a) Grad-CAM visualization for 6 different organs. We randomly draw 10 images for each organ and present the original grayscale image and the gradient map overlaid on the image together. (b) Inner product visualization for 3 randomly sampled phecodes that include "heart". We show the top ten images with the largest inner products with the given phecode. We denote a correct pair with a green check and a red cross otherwise.

## 8 Conclusion

We introduce an RKHS framework to understand CLIP by transforming the problem of estimating a mapping $\phi$ to selecting an unknown RKHS and further reduce CLIP to a multi-task multi-kernel classifier. We prove the optimal rate of our method in a sparse additive RKHS family.

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
