## A    INCOHERENCE

To give a precise description of the relevant RKHSs, we need to introduce some measures to characterize the dependence between the spaces $\mathcal{H}_m$. The correlation of RKHSs inside the indices $I \subset \{1, ..., M\}$ can be depicted by the parameter $\kappa(I)$ as

$$\kappa(I) := \sup\left\{ k \geq 0 \,|\, k \leq \frac{\|\sum_{m \in I} f_m\|^2_{L_2(\Pi)}}{\sum_{m \in I} \|f_m\|^2_{L_2(\Pi)}}, \forall f_m \in \mathcal{H}_m (m \in I) \right\}.$$

Similarly, to describe the correlation of RKHSs between the relevant part $I$ and the irrelevant part $I^c$, we introduce $\rho(I)$ as

$$\rho(I) := \sup\left\{ \frac{\langle f_I, g_{I^c}\rangle_{L_2(\Pi)}}{\|f_I\|_{L_2(\Pi)}\|g_{I^c}\|_{L_2(\Pi)}} \,|\, f_{I_t} \in \mathcal{H}_I, g_{I^c} \in \mathcal{H}_{I^c}, f_I \neq 0, g_{I^c} \neq 0 \right\}.$$

We denote the common subset of all active kernels $I_0 = \cup_{t=1}^T I_t$ and define

$$(1 - \rho(I_0)^2)\kappa(I_0) := \max_{t=1,...,T}(1 - \rho(I_t)^2)\kappa(I_t) \quad \text{and} \quad d := |I_0|.$$

These quantities give a connection between $\sum_{t=1}^T \|f_t\|^2_{L_2(\Pi)}$ and $\sum_{t=1}^T \sum_{m \in I_0} \|f_{mt}\|^2_{L_2(\Pi)}$ as stated in the following lemma. The proof can be found Section F.

**Lemma A.1.** For all $I_t \subset \{1, ..., M\}$ and $t = 1, \ldots, T$, we have

$$\sum_{t=1}^T \|f_t\|^2_{L_2(\Pi)} \geq (1 - \rho(I_0)^2)\kappa(I_0) \sum_{t=1}^T \sum_{m \in I_0} \|f_{mt}\|^2_{L_2(\Pi)}.$$

The following assumption states that the RKHSs are not too dependent on each other.

**Assumption A.2** (Incoherence Assumption). For the truly active components $I_0$, we have

$$0 < \kappa(I_0)(1 - \rho(I_0)^2).$$

The Incoherence Assumption is commonly used in sparse additive models [34; 44; 65]. Under this assumption, we can focus on deriving the bound of the relevant part $I_0$ instead of all $M$ components.

## B    BLOCK COORDINATE GRADIENT DESCENT

---

**Algorithm 1** Block coordinate gradient descent for multi-task multi-kernel logistic regression

---

**Input:** Data $(\boldsymbol{x}_{ti}, y_{ti})$, regularization parameters $\lambda_1, \lambda_2, \lambda_3$, Hessian parameter $h$, line search parameters.
Initialize $\boldsymbol{\gamma}$.
**for** iteration $= 1, 2, \ldots$ **do**
  **for** $m = 1, 2, \ldots, M$ **do**
    Compute $\boldsymbol{d}_m = \operatorname{argmin} M_m(\boldsymbol{d}_m)$ in (B.3).
    **if** $\boldsymbol{d}_m \neq 0$ **then**
      Compute step size $\tau$ with backtracking line search.
      Update $\boldsymbol{\gamma}_m \leftarrow \boldsymbol{\gamma}_m + \tau \boldsymbol{d}_m$.
    **end if**
  **end for**
**end for**
**Output:** Estimator $\widehat{f}_{mt} = \sum_{i=1}^n \alpha_{mti} k_m(\boldsymbol{x}_{ti}, \cdot)$.

---

The loss function is given as

$$L(f) = \sum_{t=1}^T \sum_{i=1}^n \log\left(1 + \exp\left(-y_{ti} f_t(\boldsymbol{x}_{ti})\right)\right) + \lambda_1 \sum_{m=1}^M \sqrt{\sum_{t=1}^T \|f_{mt}\|^2_n}$$

$$+ \lambda_2 \sum_{m=1}^M \sqrt{\sum_{t=1}^T \|f_{mt}\|^2_{\mathcal{H}_m}} + \lambda_3 \sum_{m=1}^M \sum_{t=1}^T \|f_{mt}\|^2_{\mathcal{H}_m}. \tag{B.1}$$

Let $\{(\boldsymbol{x}_{ti}, y_{ti})\}_{i=1}^n$ be $n$ independent random samples of $(\boldsymbol{X}_t, Y_t)$ distributed according to (3.1). By the representer theorem [28] and the kernel trick, we can express the solution $\widehat{f}_{mt}$ as a linear combination of kernels: $\widehat{f}_{mt}(\boldsymbol{x}) = \sum_{i=1}^n \alpha_{mi} k_m(\boldsymbol{x}_{ti}, \boldsymbol{x})$. Thus, (3.2) can be rewritten as

$$L(\boldsymbol{\alpha}) = \frac{1}{n} \sum_{t=1}^T \sum_{i=1}^n \log(1 + \exp(-y_{ti} \sum_{m=1}^M \boldsymbol{k}_{mti}^T \boldsymbol{\alpha}_{mt})) + \lambda_1 \sum_{m=1}^M \sqrt{\frac{1}{n} \sum_{t=1}^T \boldsymbol{\alpha}_{mt}^T \boldsymbol{K}_{mt} \boldsymbol{K}_{mt} \boldsymbol{\alpha}_{mt}}$$

$$+ \lambda_2 \sum_{m=1}^M \sqrt{\sum_{t=1}^T \boldsymbol{\alpha}_{mt}^T \boldsymbol{K}_{mt} \boldsymbol{\alpha}_{mt}} + \lambda_3 \sum_{t=1}^T \sum_{m=1}^M \boldsymbol{\alpha}_{mt}^T \boldsymbol{K}_{mt} \boldsymbol{\alpha}_{mt}, \tag{B.2}$$

where $\boldsymbol{K}_{mt} = (k_m(\boldsymbol{x}_{ti}, \boldsymbol{x}_{tj}))_{i,j}$ is the Gram matrix, $\boldsymbol{k}_{mti}$ is the $i$th column vector for the Gram matrix $\boldsymbol{K}_{mt} = (\boldsymbol{k}_{mt1}, \boldsymbol{k}_{mt2}, \ldots, \boldsymbol{k}_{mtn})$, and $\boldsymbol{\alpha}_{mt} = (\alpha_{mt1}, \alpha_{mt2}, \ldots, \alpha_{mtn})^T \in \mathbb{R}^n$ is the coefficient vector for the kernel $m$ and task $t$. Next we introduce some extra notations for further simplification. Denote $\boldsymbol{\gamma}_m = (\boldsymbol{\alpha}_{m1}^T, \boldsymbol{\alpha}_{m2}^T, \ldots, \boldsymbol{\alpha}_{mT}^T)^T \in \mathbb{R}^{nT}$, where we stack $\boldsymbol{\alpha}_{mt}$ together for $T$ tasks. We also write $\boldsymbol{W}_m = \text{diag}(\boldsymbol{K}_{m1}, \boldsymbol{K}_{m2}, \ldots, \boldsymbol{K}_{mT}) \in \mathbb{R}^{nT \times nT}$, where we combine Gram matrices by constructing a diagonal block matrix. Using the above notations, the objective function in (B.2) can be written as

$$L(\boldsymbol{\gamma}) = \frac{1}{n} \sum_{t=1}^T \sum_{i=1}^n \log(1 + \exp(-y_{ti} \sum_{m=1}^M \boldsymbol{k}_{mti}^T \boldsymbol{\alpha}_{mt})) + \lambda_1 \sum_{m=1}^M \sqrt{\frac{1}{n} \boldsymbol{\gamma}_m^T \boldsymbol{W}_m \boldsymbol{W}_m \boldsymbol{\gamma}_m}$$

$$+ \lambda_2 \sum_{m=1}^M \sqrt{\boldsymbol{\gamma}_m^T \boldsymbol{W}_m \boldsymbol{\gamma}_m} + \lambda_3 \sum_{m=1}^M \boldsymbol{\gamma}_m^T \boldsymbol{W}_m \boldsymbol{\gamma}_m.$$

The objective function is now transformed into a typical Lasso-type loss. One can use the SOCP algorithm [3] or the alternative direction method of multipliers [6] to solve the problem. However, we accelerate the optimization by exploiting the special structure induced by multiple kernels. More specifically, we perceive each kernel as a block and consider the block coordinate gradient descent (BCGD) algorithm proposed in Meier et al. [43]. The key idea of BCGD is to combine a second-order approximation of the log-likelihood with line search and update each penalized kernel in a block-wise manner. We denote $\ell(\boldsymbol{\gamma}) = -\frac{1}{n} \sum_{t=1}^T \sum_{i=1}^n \log(1 + \exp(-y_{ti} \sum_{m=1}^M \boldsymbol{k}_{mti}^T \boldsymbol{\alpha}_{mt}))$ as the log-likelihood and $\nabla \ell(\boldsymbol{\gamma})$ as the gradient of $\ell(\boldsymbol{\gamma})$ with regards to $\boldsymbol{\gamma}$. We write $\boldsymbol{d} \in \mathbb{R}^{nT}$ as the search direction and denote $\boldsymbol{d}_m = (d_{m1}, d_{m2}, \ldots, d_{mT})^T$. Thus, by applying a quadratic approximation to $\ell(\boldsymbol{\gamma})$, we can approximate $L(\boldsymbol{\gamma} + \boldsymbol{d})$ with $M(\boldsymbol{d})$ given as

$$M(\boldsymbol{d}) = -\ell(\boldsymbol{\gamma}) - \boldsymbol{d}^T \nabla \ell(\boldsymbol{\gamma}) + \boldsymbol{d}^T \boldsymbol{H} \boldsymbol{d} + \lambda_1 \sum_{m=1}^M \sqrt{\frac{1}{n} (\boldsymbol{\gamma}_m + \boldsymbol{d}_m)^T \boldsymbol{W}_m \boldsymbol{W}_m (\boldsymbol{\gamma}_m + \boldsymbol{d}_m)}$$

$$+ \lambda_2 \sum_{m=1}^M \sqrt{(\boldsymbol{\gamma}_m + \boldsymbol{d}_m)^T \boldsymbol{W}_m (\boldsymbol{\gamma}_m + \boldsymbol{d}_m)} + \lambda_3 \sum_{m=1}^M (\boldsymbol{\gamma}_m + \boldsymbol{d}_m)^T \boldsymbol{W}_m (\boldsymbol{\gamma}_m + \boldsymbol{d}_m),$$

where $\boldsymbol{H} \in \mathbb{R}^{nTM \times nTM}$ is a suitable matrix to replace the Hessian of the log-likelihood. Now we consider the minimization of $M(\boldsymbol{d})$ regarding each kernel $m$ by setting $\boldsymbol{d}_j = 0$ for $j \neq m$. This means that we only update one kernel $m$ of all tasks at each time. Moreover, we consider $\boldsymbol{H} = -h \odot \text{diag}(\boldsymbol{W}_1, \boldsymbol{W}_2, \ldots, \boldsymbol{W}_M)$ for some constant $h > 0$. And we need to minimize

$$M_m(\boldsymbol{d}_m) = -\boldsymbol{d}_m^T \nabla \ell(\boldsymbol{\gamma}_m) + \frac{h}{2} \boldsymbol{d}_m^T \boldsymbol{W}_m \boldsymbol{d}_m + \lambda_1 \sqrt{\frac{1}{n} (\boldsymbol{\gamma}_m + \boldsymbol{d}_m)^T \boldsymbol{W}_m \boldsymbol{W}_m (\boldsymbol{\gamma}_m + \boldsymbol{d}_m)}$$

$$+ \lambda_2 \sqrt{(\boldsymbol{\gamma}_m + \boldsymbol{d}_m)^T \boldsymbol{W}_m (\boldsymbol{\gamma}_m + \boldsymbol{d}_m)} + \lambda_3 (\boldsymbol{\gamma}_m + \boldsymbol{d}_m)^T \boldsymbol{W}_m (\boldsymbol{\gamma}_m + \boldsymbol{d}_m) \tag{B.3}$$

and perform a backtracking line search to obtain a proper step size $\tau$. We derive the closed-form solution to (B.3) when $\lambda_1 = 0$

$$M_m(\boldsymbol{d}_m) = -\boldsymbol{d}_m^T \nabla \ell(\boldsymbol{\gamma}_m) + \frac{h}{2} \boldsymbol{d}_m^T \boldsymbol{W}_m \boldsymbol{d}_m + \lambda_2 \sqrt{(\boldsymbol{\gamma}_m + \boldsymbol{d}_m)^T \boldsymbol{W}_m (\boldsymbol{\gamma}_m + \boldsymbol{d}_m)}$$

$$+ \lambda_3 (\boldsymbol{\gamma}_m + \boldsymbol{d}_m)^T \boldsymbol{W}_m (\boldsymbol{\gamma}_m + \boldsymbol{d}_m).$$

Let $\boldsymbol{t}_m = \boldsymbol{\gamma}_m + \boldsymbol{d}_m$, we complete the square and remove the constant

$$M_m(\boldsymbol{t}_m) = -\boldsymbol{t}_m^T(\nabla\ell(\boldsymbol{\gamma}_m) + h\boldsymbol{W}_m\boldsymbol{\gamma}_m) + (\frac{h}{2} + \lambda_3)\boldsymbol{t}_m^T\boldsymbol{W}_m\boldsymbol{t}_m + \lambda_2\sqrt{\boldsymbol{t}_m^T\boldsymbol{W}_m\boldsymbol{t}_m}.$$

Let $\sqrt{\boldsymbol{W}_m}\boldsymbol{t}_m = \boldsymbol{\beta}_m$, we have

$$M_m(\boldsymbol{\beta}_m) = -\boldsymbol{\beta}_m^T(\sqrt{\boldsymbol{W}_m}^{-1}\nabla\ell(\boldsymbol{\gamma}_m) + h\sqrt{\boldsymbol{W}_m}\boldsymbol{\gamma}_m) + (\frac{h}{2} + \lambda_3)\boldsymbol{\beta}_m^T\boldsymbol{\beta}_m + \lambda_2\sqrt{\boldsymbol{\beta}_m^T\boldsymbol{\beta}_m}.$$

Therefore, the derivative is given as

$$M_m'(\boldsymbol{\beta}_m) = -(\sqrt{\boldsymbol{W}_m}^{-1}\nabla\ell(\boldsymbol{\gamma}_m) + h\sqrt{\boldsymbol{W}_m}\boldsymbol{\gamma}_m) + (h + 2\lambda_3)\boldsymbol{\beta}_m + \lambda_2\frac{\boldsymbol{\beta}_m}{\sqrt{\boldsymbol{\beta}_m^T\boldsymbol{\beta}_m}}.$$

Thus, by setting $M_m'(\boldsymbol{\beta}_m) = 0$ and the relation $\sqrt{\boldsymbol{W}_m}\boldsymbol{t}_m = \boldsymbol{\beta}_m$ and $\boldsymbol{t}_m = \boldsymbol{\gamma}_m + \boldsymbol{d}_m$, we have

$$\boldsymbol{d}_m = \frac{1}{h + 2\lambda_3}\Bigg[(\boldsymbol{W}_m^{-1}\nabla\ell(\boldsymbol{\gamma}_m) + h\boldsymbol{\gamma}_m) \tag{B.4}$$

$$- \lambda_2\frac{\boldsymbol{W}_m^{-1}\nabla\ell(\boldsymbol{\gamma}_m) + h\boldsymbol{\gamma}_m}{\sqrt{\nabla\ell(\boldsymbol{\gamma}_m)^T\boldsymbol{W}_m^{-1}\nabla\ell(\boldsymbol{\gamma}_m) + 2h\boldsymbol{\gamma}_m^T\nabla\ell(\boldsymbol{\gamma}_m) + h^2\boldsymbol{\gamma}_m^T\boldsymbol{W}_m\boldsymbol{\gamma}_m}}\Bigg] - \boldsymbol{\gamma}_m.$$

As a result, if $\sqrt{\nabla\ell(\boldsymbol{\gamma}_m)^T\boldsymbol{W}_m^{-1}\nabla\ell(\boldsymbol{\gamma}_m) + 2h\boldsymbol{\gamma}_m^T\nabla\ell(\boldsymbol{\gamma}_m) + h^2\boldsymbol{\gamma}_m^T\boldsymbol{W}_m\boldsymbol{\gamma}_m} \le \lambda_2$, $M_m(\boldsymbol{d}_m)$ is maximized by $\boldsymbol{d}_m = \boldsymbol{\gamma}_m$. Otherwise, it is maximized by (B.4). We have made PyTorch code publicly available[1].

## C  CONFIGURATION OF SIMULATION

We consider the following models:

- **MTMK mix.** We consider our multi-task multi-kernel model with the elastic-net type penalty.
- **MTMK L1.** This is our multi-task multi-kernel model with only $L_1$ penalty. We set $\lambda_3 = 0$ to induce more sparsity than the MTMK mix model.
- **MTMK L2.** With only $L_2$ penalty, this model is simply a group of $T$ multi-kernel logistic regression with the same $L_2$ regularization parameter for each task. Thus, it is essentially different from our proposed multi-task model.
- **STMK.** In this model, we consider $T$ sparse single-task multi-kernel logistic regression. Each task can generate a distinct sparsity pattern, which might not be favorable given that all tasks share the same subset of truly active kernels.
- **MTSK.** We consider a multi-task single-kernel model with $L_2$ penalty. Specifically, we use a gaussian kernel $K(x, y) = \exp(\|x - y\|^2)$ for its promising approximating ability for a fair comparison.
- **STSK.** This is a single-task single-kernel model with $L_2$ penalty. We use the same kernel as in MTSK. Since the loss function for each task is convex, STSK is theoretically the same as MTSK, but a minor numerical difference might be observed.
- **ORACLE.** Only true kernels are given as input in ORACLE. This model indicates the best possible performance that can be achieved by the models listed above. We use a multi-kernel model with a small $L_2$ penalty for numerical stability.

We perform grid search in $\{0.005, 0.01, \ldots, 0.5\}$ to tune hyperparameters $\lambda_2$ and $\lambda_3$ for all models except MTMK mix, for which $\{0.001, 0.005, 0.01, \ldots, 0.5\}$ is used. Because the true function is unknown in a real-world scenario and the mean squared error cannot be computed, we use the likelihood function to select the best hyperparameters. To form our candidate kernel pool, we manually construct 22 candidate kernels, including Gaussian kernels with $\gamma = \{10^{-5}, 10^{-4}, \ldots, 10^0\}$, sigmoid

---

[1] https://anonymous.4open.science/r/CLIP_as_MTMK-6786/

kernels with $\gamma = \{10^{-5}, 10^{-4}, \ldots, 10^{0}\}$, and polynomial kernels with a degree of $\{1, 2, \ldots, 10\}$. In this experiment, we investigate three distinct scenarios to evaluate the performance of our multi-task model. The first task set comprises a combination of $X, X^3$, and $X^5$, where one of the coefficients in each task is assigned a value of $0.1$. This scenario particularly favors multi-task models as it poses a challenge for single-task models to accurately identify the correct kernels, given the presence of a kernel with a small weight in each task. The second task set is a combination of $X, X^2$ and $X^3$, where not all true kernels appear in each task. It is more challenging for our multi-task model to learn the sparsity pattern compared to the first task set because the correlation of each task becomes a bit weaker. The third task set includes a combination of $X, X^3$ and $X^5$, a simple scenario where all tasks share the same true kernels. But models like STMK can demonstrate competitive performance in this scenario given that each task is easy. Therefore, our aim in this scenario is to evaluate whether the joint estimation ability of our MTMK model can still provide better performance compared to other models. Detailed configuration is summarized in Table 3.

Table 3: Task sets configuration.

| Task Sets | Tasks | |
|---|---|---|
| Task Set 1 | $f_1(X) = X + X^3 + 0.1X^5$ | $f_4(X) = -X - 0.1X^3 - X^5$ |
| | $f_2(X) = -X - X^3 - 0.1X^5$ | $f_5(X) = 0.1X + X^3 + X^5$ |
| | $f_3(X) = X + 0.1X^3 + X^5$ | $f_6(X) = -0.1X - X^3 - X^5$ |
| Task Set 2 | $f_1(X) = X^3$ | $f_4(X) = -0.1X + X^2$ |
| | $f_2(X) = X^2$ | $f_5(X) = -X + 3X^2$ |
| | $f_3(X) = X$ | $f_6(X) = 2X^2 - X^3$ |
| Task Set 3 | $f_1(X) = 5X + -1.25X^3 - 1.25X^5$ | $f_4(X) = -5X + 1.25X^3 + 1.25X^5$ |
| | $f_2(X) = -1.25X + 1.25X^3 - 2.5X^5$ | $f_5(X) = 1.25X - 1.25X^3 + 2.5X^5$ |
| | $f_3(X) = -1.25X - 1.25X^3 + 6X^5$ | $f_6(X) = 1.25X + 1.25X^3 - 6X^5$ |

## D MNIST

MNIST is a widely recognized handwritten digits database consisting of 60,000 training images and 10,000 testing images. Each image is a grayscale $28 \times 28$ representation of a handwritten digit $(0 - 9)$, labeled with its corresponding number. We randomly select 200 samples from each class in the training images for our experiments and adopt a one-vs-rest strategy. For each class, we randomly select 100 images from each corresponding category as positive samples and draw 100 negative samples from the other categories. All images are flattened into a vector of 784 dimensions after preprocessing. Consistent with prior research [38], we anticipate significant inter-task correlation in our curated multi-task dataset. Similar to our experiment in section 7.1, we perform grid search to tune all hyperparameters. We conduct a 5-fold cross-validation and compute the average AUROC. Results are summarized in Table 4.

Table 4: Average AUROC under 5-fold cross validation.

| | LR | SVM | MTSK | STMK | MTMK L1 |
|---|---|---|---|---|---|
| Task 1 | 0.9903 | 0.9934 | 0.9908 | 0.9934 | **0.9934** |
| Task 2 | 0.9975 | 0.9973 | **0.9975** | 0.9954 | 0.9960 |
| Task 3 | 0.9582 | 0.9582 | 0.9588 | 0.9784 | **0.9801** |
| Task 4 | 0.9464 | 0.9585 | 0.9475 | **0.9595** | 0.9573 |
| Task 5 | 0.9726 | 0.9685 | 0.9698 | **0.9895** | 0.9878 |
| Task 6 | 0.9613 | **0.9790** | 0.9631 | 0.9736 | 0.9752 |
| Task 7 | 0.9868 | 0.9886 | 0.9826 | 0.9987 | **0.9987** |
| Task 8 | 0.9740 | 0.9713 | 0.9729 | **0.9909** | 0.9903 |
| Task 9 | 0.9556 | **0.9685** | 0.9539 | 0.9492 | 0.9527 |
| Task 10 | 0.8947 | 0.8224 | 0.8953 | 0.9678 | **0.9684** |
| Average | 0.9637 | 0.9605 | 0.9632 | 0.9796 | **0.9800** |

# E   PROOF OF MAIN RESULT

We start from the fact that $\widehat{f}$ minimizes the Equation (3.2)

$$L(\widehat{f}) \leq L(f^*).$$

We define the logistic loss as

$$\ell(f_{mt}) := \sum_{t=1}^{T} \sum_{i=1}^{n} \log\left(1 + \exp\left(-y_{ti} \sum_{m=1}^{M} f_{mt}(\boldsymbol{x}_{ti})\right)\right),$$

and the gradient of the logistic loss as

$$\epsilon_{ti} := \frac{-y_{ti} \exp(-y_{ti} \sum_{m=1}^{M} f_{mt}^*(\boldsymbol{x}_{ti}))}{1 + \exp(-y_{ti} \sum_{m=1}^{M} f_{mt}^*(\boldsymbol{x}_{ti}))}.$$

The following lemma is the first step to our main results.

**Lemma E.1.** Define $C_\alpha := \frac{\exp(-C)}{2(1+\exp(C))^2}$. Under the Union Bound Assumption, we have

$$\ell(\widehat{f}_{mt}) - \ell(f_{mt}^*) \geq \sum_{i=1}^{n} \sum_{t=1}^{T} \sum_{m=1}^{M} \epsilon_{ti}(\widehat{f}_{mt}(\boldsymbol{x}_{ti}) - f_{mt}^*(\boldsymbol{x}_{ti})) + C_\alpha \sum_{i=1}^{n} \sum_{t=1}^{T} (\widehat{f}_t(\boldsymbol{x}_{ti}) - f_t^*(\boldsymbol{x}_{ti}))^2.$$

*Proof of Lemma E.1.* We write $\ell'(f_{mt})_i$ to denote the derivative of $\ell(f_{mt})$ with respect to $f_{mt}(\boldsymbol{x}_{ti})$. By definition, we obtain

$$\ell'(f_{mt})_i = \frac{-y_{ti} \exp(-y_{ti} \sum_{m=1}^{M} f_{mt}(\boldsymbol{x}_{ti}))}{1 + \exp(-y_{ti} \sum_{m=1}^{M} f_{mt}(\boldsymbol{x}_{ti}))} \quad \text{and} \quad \ell''(f_{mt})_i = \frac{y_{ti}^2 \exp(-y_{ti} \sum_{m=1}^{M} f_{mt}(\boldsymbol{x}_{ti}))}{(1 + \exp(-y_{ti} \sum_{m=1}^{M} f_{mt}(\boldsymbol{x}_{ti})))^2}.$$

By the Union Bound Assumption, we have

$$\ell''(f_{mt}^*)_i \geq \frac{\exp(-C_\beta)}{2(1 + \exp(C_\beta))^2}.$$

Thus, using the fact that $\sum_{m=1}^{M} \|\widehat{f}_{mt}\|_{\mathcal{H}_m} \leq C$ for some constant $C > 0$, we get the assertion by combining Taylor expansion and the above inequality. □

Using Lemma E.1, through simple calculation, we obtain

$$\frac{1}{n} \sum_{t=1}^{T} \sum_{i=1}^{n} \epsilon_{ti}(\widehat{f}_t(\boldsymbol{x}_{ti}) - f_t^*(\boldsymbol{x}_{ti})) + \frac{\exp(-C)}{2(1 + \exp(C))^2} \frac{1}{n} \sum_{t=1}^{T} \sum_{i=1}^{n} (\widehat{f}_t(\boldsymbol{x}_{ti}) - f_t^*(\boldsymbol{x}_{ti}))^2$$

$$+ \lambda_1 \sum_{m=1}^{M} \sqrt{\sum_{t=1}^{T} \|\widehat{f}_{mt}\|_n^2} + \lambda_2 \sum_{m=1}^{M} \sqrt{\sum_{t=1}^{T} \|\widehat{f}_{mt}\|_{\mathcal{H}_m}^2} + \lambda_3 \sum_{m=1}^{M} \sum_{t=1}^{T} \|\widehat{f}_{mt}\|_{\mathcal{H}_m}^2$$

$$\leq \lambda_1 \sum_{m \in I_0} \sqrt{\sum_{t=1}^{T} \|f_{mt}^*\|_n^2} + \lambda_2 \sum_{m \in I_0} \sqrt{\sum_{t=1}^{T} \|f_{mt}^*\|_{\mathcal{H}_m}^2} + \lambda_3 \sum_{m \in I_0} \sum_{t=1}^{T} \|f_{mt}^*\|_{\mathcal{H}_m}^2. \tag{E.1}$$

Recall the definition of $C_\alpha$, we get

$$\sum_{t=1}^{T} \|\widehat{f}_t - f_t^*\|_{L_2(\Pi)}^2 + \frac{\lambda_1}{C_\alpha} \sum_{m=1}^{M} \sqrt{\sum_{t=1}^{T} \|\widehat{f}_{mt}\|_n^2} + \frac{\lambda_2}{C_\alpha} \sum_{m=1}^{M} \sqrt{\sum_{t=1}^{T} \|\widehat{f}_{mt}\|_{\mathcal{H}_m}^2} + \frac{\lambda_3}{C_\alpha} \sum_{m=1}^{M} \sum_{t=1}^{T} \|\widehat{f}_{mt}\|_{\mathcal{H}_m}^2$$

$$\leq \sum_{t=1}^{T} \left( \|\widehat{f}_t - f_t^*\|_{L_2(\Pi)}^2 - \|\widehat{f}_t - f_t^*\|_n^2 \right) + \left| \frac{1}{n} \sum_{t=1}^{T} \sum_{i=1}^{n} \frac{1}{C_\alpha} \epsilon_{ti}(\widehat{f}_t(\boldsymbol{x}_{ti}) - f_t^*(\boldsymbol{x}_{ti})) \right|$$

$$+ \frac{\lambda_1}{C_\alpha} \sum_{m \in I_0} \sqrt{\sum_{t=1}^{T} \|f_{mt}^*\|_n^2} + \frac{\lambda_2}{C_\alpha} \sum_{m \in I_0} \sqrt{\sum_{t=1}^{T} \|f_{mt}^*\|_{\mathcal{H}_m}^2} + \frac{\lambda_3}{C_\alpha} \sum_{m \in I_0} \sum_{t=1}^{T} \|f_{mt}^*\|_{\mathcal{H}_m}^2.$$

Here we need to give an upper probability bound for the first two terms on the RHS with some type of group Lasso norm $\sum_{m=1}^{M} \sqrt{\sum_{t=1}^{T} \|f_{mt}\|^2}$. More specifically, we consider $\sum_{m=1}^{M} \left( \sqrt{\sum_{t=1}^{T} \|f_{mt}\|_{L_2(\Pi)}^2} + \lambda^{\frac{1}{2}} \sqrt{\sum_{t=1}^{T} \|f_{mt}\|_{\mathcal{H}_m}^2} \right)$. To provide more intuition on the events to be introduced, we need the bound of $|\frac{1}{n} \sum_{m=1}^{M} \sum_{t=1}^{T} \sum_{i=1}^{n} \epsilon_{ti} f_{mt}(\boldsymbol{x}_{ti})|$ with a rate of $\sqrt{\frac{1}{n^{1/(1+s)}} (T + \sqrt{T} \log M)}$ and $\sum_{t=1}^{T} \left| \|\sum_{m=1}^{M} f_{mt}\|_n^2 - \|\sum_{m=1}^{M} f_{mt}\|_{L_2(\Pi)}^2 \right|$ with a rate faster than $\frac{1}{n^{(1-s)/2(1+s)}} + \frac{\log M}{\sqrt{n}}$, both with the norm mentioned above. Now we include more notations to strictly describe the two random events. Define $\xi_n, \omega_n$ as

$$\xi_n := \max \left( \frac{\lambda^{-\frac{s}{2}}}{\sqrt{n}}, \frac{\lambda^{-\frac{1}{2}}}{n^{\frac{1}{1+s}}}, \sqrt{\frac{\log M}{n}} \right),$$

$$\omega_n := \max \left( \frac{1}{n}, \sqrt{\frac{6}{n^{2/(1+s)}} + \frac{C_1^4}{n^{(3+s)/(1+s)} \lambda^{2s}}}, \frac{\lambda^{-s}}{n}, \frac{\lambda^{-1}}{n^{\frac{2}{1+s}}} \right),$$

where $c_s$ is a constant depending on $s$ appeared in Proposition G.1. In addition, $\zeta_{nT}$ is used in giving probability:

$$\zeta_{nT}(r, \lambda) := \min \left( \frac{r^2 \log M}{n \xi_n(\lambda)^4 c_s^2}, \frac{r}{\xi_n(\lambda)^2 c_s} \right) - \log T.$$

We consider the following two random events $\mathcal{E}_1(u)$ and $\mathcal{E}_2(r)$:

$$\mathcal{E}_1(u) = \left\{ \left| \frac{1}{n} \sum_{t=1}^{T} \sum_{i=1}^{n} \epsilon_{ti} f_{mt}(\boldsymbol{x}_{ti}) \right| \leq \sqrt{c_\nu \omega_n (T + \sqrt{T} \log M) u} \right.$$
$$\left. \left( \sqrt{\sum_{t=1}^{T} \|f_{mt}\|_{L_2(\Pi)}^2} + \lambda^{\frac{1}{2}} \sqrt{\sum_{t=1}^{T} \|f_{mt}\|_{\mathcal{H}_m}^2} \right), \forall f_{mt} \in \mathcal{H}_m, \forall m = 1, \ldots, M \right\},$$

$$\mathcal{E}_2(r) = \left\{ \sum_{t=1}^{T} \left| \left\| \sum_{m=1}^{M} f_{mt} \right\|_n^2 - \left\| \sum_{m=1}^{M} f_{mt} \right\|_{L_2(\Pi)}^2 \right| \leq \max(c_s \sqrt{n} \xi_n^2, r) \right.$$
$$\left. \left[ \sum_{m=1}^{M} \left( \sqrt{\sum_{t=1}^{T} \left\| f_{mt} \right\|_{L_2(\Pi)}^2} + \lambda^{\frac{1}{2}} \sqrt{\sum_{t=1}^{T} \left\| f_{mt} \right\|_{\mathcal{H}_m}^2} \right) \right]^2, \forall f_{mt} \in \mathcal{H}_m, \forall m = 1, \ldots, M \right\}.$$

Then the following theorems indicate these random events hold with high probability.

**Theorem E.2.** Under the Bounded Kernel Assumption, the Spectral Assumption and the Sup-norm Assumption, we have for all $\lambda > 0$ and all $u \geq 1$

$$\mathbb{P}\left( \mathcal{E}_1(u) \right) \geq 1 - \exp(-u) - \frac{M}{n^{s/(1+s)}} \sqrt{\exp\left( -(T^{1/2} \wedge n^{1/2} T^{1/4}) \right)}.$$

**Theorem E.3.** Under the Spectral Assumption and the Sup-norm Assumption, we have for all $\lambda > 0$ and all $r > 0$

$$\mathbb{P}\left( \mathcal{E}_2(r) \right) \geq 1 - \exp(-\zeta_{nT}(r, \lambda)).$$

The proof can be found in section G.

The next lemma gives a bound of irrelevant components in terms of truly active components $I_0$, which is essential in our structured sparsity setting. Using this lemma and the above two events $\mathcal{E}_1(u)$ and $\mathcal{E}_2(r)$, we can show the convergence rates of the elastic-net model and the $L_1$ model.

**Lemma E.4.** Set $\lambda_1 = 4\sqrt{c_\nu \omega_n(T + \sqrt{T}\log M)u}$ and $\lambda_2 = \lambda^{\frac{1}{2}}\lambda_1$ for arbitrary $\lambda > 0$ and $\lambda_3 \geq 0$. Then for any n and r satisfying $\frac{\log M}{\sqrt{n}} \leq 1$ and $\max(c_s\sqrt{n}\xi_n^2, r) \leq \frac{1}{8}$, we have

$$
\sum_{m=1}^{M}\left(\lambda_1\sqrt{\sum_{t=1}^{T}\|\widehat{f}_{mt} - f_{mt}^*\|_{L_2(\Pi)}^2} + \lambda_2\sqrt{\sum_{t=1}^{T}\|\widehat{f}_{mt} - f_{mt}^*\|_{\mathcal{H}_m}^2}\right)
$$

$$
\leq 8\sum_{m\in I_0}\left[\lambda_1\sqrt{\sum_{t=1}^{T}\|\widehat{f}_{mt} - f_{mt}^*\|_{L_2(\Pi)}^2} + \lambda_2\sqrt{\sum_{t=1}^{T}\|\widehat{f}_{mt} - f_{mt}^*\|_{\mathcal{H}_m}^2}\right. \tag{E.2}
$$

$$
\left. + \lambda_3^{\frac{1}{2}}\sqrt{\sum_{t=1}^{T}\|f_{mt}^*\|_{\mathcal{H}_m}^2}\left(\sqrt{\sum_{t=1}^{T}\|\widehat{f}_{mt} - f_{mt}^*\|_{L_2(\Pi)}^2} + \lambda_3^{\frac{1}{2}}\sqrt{\sum_{t=1}^{T}\|\widehat{f}_{mt} - f_{mt}^*\|_{\mathcal{H}_m}^2}\right)\right]
$$

on the events $\mathcal{E}_1(u)$ and $\mathcal{E}_2(r)$.

The proof can be found in section H.

We are ready to show the convergence rates of our models.

**Theorem E.5.** Suppose all assumptions are satisfied. Let $\lambda_1 = 4\sqrt{c_\nu\omega_n(T + \sqrt{T}\log M)u}$, $\lambda_2 = \lambda^{\frac{1}{2}}\lambda_1$, and $\lambda_3 = \lambda$ or $\lambda_3 = 0$ for arbitrary $\lambda > 0$. For all $n$ and $r$ satisfying

$$
\frac{256\max(c_s\sqrt{n}\xi_n^2, r)\left(d + \frac{\lambda_3\sum_{m\in I_0}\sum_{t=1}^{T}\|f_{mt}^*\|_{\mathcal{H}_m}^2}{\lambda_1^2}\right)}{(1 - \rho(I_0)^2))\kappa(I_0)} \leq \frac{1}{8},
$$

we have

$$
\sum_{t=1}^{T}\|\widehat{f}_t - f_t^*\|_{L_2(\Pi)}^2 \leq \frac{96}{C_\alpha^2(1 - \rho(I_0)^2))\kappa(I_0)}\left(d\lambda_1^2 + \lambda_3\sum_{m\in I_0}\sum_{t=1}^{T}\|f_{mt}^*\|_{\mathcal{H}_m}^2\right),
$$

with probability at least $1 - \exp(-u) - \frac{M}{n^{s/(1+s)}}\sqrt{\exp\left(-(T^{1/2} \wedge n^{1/2}T^{1/4})\right)} - \exp\left(-\zeta_{nT}\left(\frac{1}{\widetilde{C}_1 d}, \lambda\right)\right)$ for all $u \geq 1$.

*Proof of Theorem E.5.* Notice that the assumption stated in Theorem E.5 implies $\max(c_s\sqrt{n}\xi_n^2, r) \leq \frac{1}{8}$. Thus the condition in Lemma E.4 is met. In the proof of Lemma E.4, we show that the following inequality holds on the events $\mathcal{E}_1(t)$ and $\mathcal{E}_2(r)$ in Equation (H.3) :

$$
\sum_{t=1}^{T}\|\widehat{f}_t - f_t^*\|_n^2 + \frac{1}{C_\alpha}\frac{1}{2}\sum_{m\in I_0^c}\left(\lambda_1\sqrt{\sum_{t=1}^{T}\|\widehat{f}_{mt}\|_{L_2(\Pi)}^2} + \lambda_2\sqrt{\sum_{t=1}^{T}\|\widehat{f}_{mt}\|_{\mathcal{H}_m}^2}\right)
$$

$$
\leq \sum_{m=1}^{M}\frac{1}{C_\alpha}\sqrt{c_\nu\omega_n(T + \sqrt{T}\log M)u}\left(\sqrt{\sum_{t=1}^{T}\|\widehat{f}_{mt} - f_{mt}^*\|_{L_2(\Pi)}^2} + \lambda^{\frac{1}{2}}\sqrt{\sum_{t=1}^{T}\|\widehat{f}_{mt} - f_{mt}^*\|_{\mathcal{H}_m}^2}\right)
$$

$$
+ \frac{1}{C_\alpha}\frac{3}{2}\sum_{m\in I_0}\left(\lambda_1\sqrt{\sum_{t=1}^{T}\|\widehat{f}_{mt} - f_{mt}^*\|_{L_2(\Pi)}^2} + \lambda_2\sqrt{\sum_{t=1}^{T}\|\widehat{f}_{mt} - f_{mt}^*\|_{\mathcal{H}_m}^2}\right)
$$

$$
+ \frac{\lambda_3}{C_\alpha}\sum_{m\in I_0}\sum_{t=1}^{T}(2\langle f_{mt}^*, f_{mt}^* - \widehat{f}_{mt}\rangle_{\mathcal{H}_m} - \|\widehat{f}_{mt} - f_{mt}^*\|_{\mathcal{H}_m}^2).
$$

Adding $L_2(\Pi)$ norm to both sides and using the event $\mathcal{E}_2(r)$, we have

$$
\sum_{t=1}^{T} \|\widehat{f}_t - f_t^*\|_{L_2(\Pi)}^2 + \frac{\lambda_3}{C_\alpha} \sum_{m \in I_0} \sum_{t=1}^{T} \|\widehat{f}_{mt} - f_{mt}^*\|_{\mathcal{H}_m}^2
$$

$$
\leq \max(c_s \sqrt{n} \xi_n^2, r) \underbrace{\left[ \sum_{m=1}^{M} \left( \sqrt{\sum_{t=1}^{T} \|\widehat{f}_{mt} - f_{mt}^*\|_{L_2(\Pi)}^2} + \lambda^{\frac{1}{2}} \sqrt{\sum_{t=1}^{T} \|\widehat{f}_{mt} - f_{mt}^*\|_{\mathcal{H}_m}^2} \right) \right]^2}_{\text{I}}
$$

$$
+ \underbrace{\sum_{m=1}^{M} \frac{1}{C_\alpha} \sqrt{c_\nu \omega_n (T + \sqrt{T} \log M) u} \left( \sqrt{\sum_{t=1}^{T} \|\widehat{f}_{mt} - f_{mt}^*\|_{L_2(\Pi)}^2} + \lambda^{\frac{1}{2}} \sqrt{\sum_{t=1}^{T} \|\widehat{f}_{mt} - f_{mt}^*\|_{\mathcal{H}_m}^2} \right)}_{\text{II}}
$$

$$
+ \underbrace{\frac{1}{C_\alpha} \frac{3}{2} \sum_{m \in I_0} \left( \lambda_1 \sqrt{\sum_{t=1}^{T} \|\widehat{f}_{mt} - f_{mt}^*\|_{L_2(\Pi)}^2} + \lambda_2 \sqrt{\sum_{t=1}^{T} \|\widehat{f}_{mt} - f_{mt}^*\|_{\mathcal{H}_m}^2} \right)}_{\text{III}}
$$

$$
+ \underbrace{\frac{2\lambda_3}{C_\alpha} \sum_{m \in I_0} \sum_{t=1}^{T} \langle f_{mt}^*, f_{mt}^* - \widehat{f}_{mt} \rangle_{\mathcal{H}_m}}_{\text{IV}}. \tag{E.3}
$$

Now we will bound each term on the RHS of Equation (E.3). The assumption $\lambda^{\frac{1}{2}} = \lambda_2/\lambda_1$ and Equation (E.2) yields

$$
\sum_{m=1}^{M} \left( \lambda_1 \sqrt{\sum_{t=1}^{T} \|\widehat{f}_{mt} - f_{mt}^*\|_{L_2(\Pi)}^2} + \lambda_2 \sqrt{\sum_{t=1}^{T} \|\widehat{f}_{mt} - f_{mt}^*\|_{\mathcal{H}_m}^2} \right)
$$

$$
= 8 \sum_{m \in I_0} \left( 1 + \frac{\lambda_3^{\frac{1}{2}} \sqrt{\sum_{t=1}^{T} \|f_{mt}^*\|_{\mathcal{H}_m}^2}}{\lambda_1} \right) \left( \lambda_1 \sqrt{\sum_{t=1}^{T} \|\widehat{f}_{mt} - f_{mt}^*\|_{L_2(\Pi)}^2} + \lambda_2 \sqrt{\sum_{t=1}^{T} \|\widehat{f}_{mt} - f_{mt}^*\|_{\mathcal{H}_m}^2} \right). \tag{E.4}
$$

**(Bound of the first term I)** By Equation (E.4) and $\lambda_3^{\frac{1}{2}} = \lambda^{\frac{1}{2}} = \lambda_2/\lambda_1$, we obtain the upper bound of the first term of the RHS of Equation (E.3) as

$$
\text{I} \leq 64 \max(c_s \sqrt{n} \xi_n^2, r) \left[ \sum_{m \in I_0} \left( 1 + \frac{\lambda_3^{\frac{1}{2}} \sqrt{\sum_{t=1}^{T} \|f_{mt}^*\|_{\mathcal{H}_m}^2}}{\lambda_1} \right) \right.
$$

$$
\left. \cdot \left( \sqrt{\sum_{t=1}^{T} \|\widehat{f}_{mt} - f_{mt}^*\|_{L_2(\Pi)}^2} + \lambda^{\frac{1}{2}} \sqrt{\sum_{t=1}^{T} \|\widehat{f}_{mt} - f_{mt}^*\|_{\mathcal{H}_m}^2} \right) \right]^2. \tag{E.5}
$$

By the Cauchy-Schwarz Inequality and the relation $(x + y)^2 \leq 2(x^2 + y^2)$, we have

$$
\left[ \sum_{m \in I_0} \left( 1 + \frac{\lambda_3^{\frac{1}{2}} \sqrt{\sum_{t=1}^{T} \|f_{mt}^*\|_{\mathcal{H}_m}^2}}{\lambda_1} \right) \left( \sqrt{\sum_{t=1}^{T} \|\widehat{f}_{mt} - f_{mt}^*\|_{L_2(\Pi)}^2} + \lambda^{\frac{1}{2}} \sqrt{\sum_{t=1}^{T} \|\widehat{f}_{mt} - f_{mt}^*\|_{\mathcal{H}_m}^2} \right) \right]^2
$$

$$
\leq \sum_{m \in I_0} \left( 1 + \frac{\lambda_3^{\frac{1}{2}} \sqrt{\sum_{t=1}^{T} \|f_{mt}^*\|_{\mathcal{H}_m}^2}}{\lambda_1} \right)^2 \sum_{m \in I_0} \left( \sqrt{\sum_{t=1}^{T} \|\widehat{f}_{mt} - f_{mt}^*\|_{L_2(\Pi)}^2} + \lambda^{\frac{1}{2}} \sqrt{\sum_{t=1}^{T} \|\widehat{f}_{mt} - f_{mt}^*\|_{\mathcal{H}_m}^2} \right)^2
$$

$$
\leq 4 \sum_{m \in I_0} \left( 1 + \frac{\lambda_3 \sum_{t=1}^{T} \|f_{mt}^*\|_{\mathcal{H}_m}^2}{\lambda_1^2} \right) \sum_{m \in I_0} \sum_{t=1}^{T} \left( \|\widehat{f}_{mt} - f_{mt}^*\|_{L_2(\Pi)}^2 + \lambda \|\widehat{f}_{mt} - f_{mt}^*\|_{\mathcal{H}_m}^2 \right).
$$

Thus the RHS of (E.5) can be further bounded as

$$\mathrm{I} \le 256 \max(c_s \sqrt{n}\xi_n^2, r)\left( d + \frac{\lambda_3 \sum_{m \in I_0} \sum_{t=1}^T \|f_{mt}^*\|_{\mathcal{H}_m}^2}{\lambda_1^2} \right)$$

$$\cdot \sum_{m \in I_0} \sum_{t=1}^T \left( \|\widehat{f}_{mt} - f_{mt}^*\|_{L_2(\Pi)}^2 + \lambda \|\widehat{f}_{mt} - f_{mt}^*\|_{\mathcal{H}_m}^2 \right).$$

By the assumption $256 \max(c_s \sqrt{n}\xi_n^2, r)\left( d + \frac{\lambda_3 \sum_{m \in I_0} \sum_{t=1}^T \|f_{mt}^*\|_{\mathcal{H}_m}^2}{\lambda_1^2} \right)/(1 - \rho(I_0)^2))\kappa(I_0) \le \frac{1}{8}$, we can further bound the above inequality as

$$\mathrm{I} \le \frac{1}{8} \sum_{t=1}^T \left( \|\widehat{f}_t - f_t^*\|_{L_2(\Pi)}^2 + \sum_{m \in I_0} \lambda_3 \|\widehat{f}_{mt} - f_{mt}^*\|_{\mathcal{H}_m}^2 \right), \tag{E.6}$$

where we use Lemma A.1.

**(Bound of the second term II)** By Equation (E.4) and the relation $\lambda_1 = 4\sqrt{c_\nu \omega_n (T + \sqrt{T} \log M)u}$, we have

$$\mathrm{II} \le \frac{2\lambda_1}{C_\alpha} \sum_{m \in I_0} \left( 1 + \frac{\lambda_3^{\frac{1}{2}} \sqrt{\sum_{t=1}^T \|f_{mt}^*\|_{\mathcal{H}_m}^2}}{\lambda_1} \right) \left( \sqrt{\sum_{t=1}^T \|\widehat{f}_{mt} - f_{mt}^*\|_{L_2(\Pi)}^2} + \lambda^{\frac{1}{2}} \sqrt{\sum_{t=1}^T \|\widehat{f}_{mt} - f_{mt}^*\|_{\mathcal{H}_m}^2} \right)$$

$$\le \sum_{m \in I_0} \frac{16}{C_\alpha^2 (1 - \rho(I_0)^2))\kappa(I_0)} \left( \lambda_1 + \lambda_3^{\frac{1}{2}} \sqrt{\sum_{t=1}^T \|f_{mt}^*\|_{\mathcal{H}_m}^2} \right)^2$$

$$+ \sum_{m \in I_0} \frac{(1 - \rho(I_0)^2))\kappa(I_0)}{16} \left( \sqrt{\sum_{t=1}^T \|\widehat{f}_{mt} - f_{mt}^*\|_{L_2(\Pi)}^2} + \lambda^{\frac{1}{2}} \sqrt{\sum_{t=1}^T \|\widehat{f}_{mt} - f_{mt}^*\|_{\mathcal{H}_m}^2} \right)^2.$$

Using Lemma A.1 and $(x + y)^2 \le 2(x^2 + y^2)$, the RHS of the above inequality can be further bounded as

$$\mathrm{II} \le \frac{32}{C_\alpha^2 (1 - \rho(I_0)^2))\kappa(I_0)} \left( d\lambda_1^2 + \lambda_3 \sum_{m \in I_0} \sum_{t=1}^T \|f_{mt}^*\|_{\mathcal{H}_m}^2 \right)$$

$$+ \frac{1}{8} \sum_{t=1}^T \left( \|\widehat{f}_t - f_t^*\|_{L_2(\Pi)}^2 + \sum_{m \in I_0} \lambda \|\widehat{f}_{mt} - f_{mt}^*\|_{\mathcal{H}_m}^2 \right). \tag{E.7}$$

**(Bound of the third term III)** Similarly, using $2xy \le x^2 + y^2$, $(x + y)^2 \le 2(x^2 + y^2)$, and Lemma A.1, we have

$$\mathrm{III} \le \sum_{m \in I_0} \frac{(1 - \rho(I_0)^2))\kappa(I_0)}{16} \left( \sqrt{\sum_{t=1}^T \|\widehat{f}_{mt} - f_{mt}^*\|_{L_2(\Pi)}^2} + \lambda^{\frac{1}{2}} \sqrt{\sum_{t=1}^T \|\widehat{f}_{mt} - f_{mt}^*\|_{\mathcal{H}_m}^2} \right)^2$$

$$+ \frac{9d\lambda_1^2}{C_\alpha^2 (1 - \rho(I_0)^2))\kappa(I_0)}$$

$$\le \frac{9d\lambda_1^2}{C_\alpha^2 (1 - \rho(I_0)^2))\kappa(I_0)} + \frac{1}{8} \sum_{t=1}^T \left( \|\widehat{f}_t - f_t^*\|_{L_2(\Pi)}^2 + \sum_{m \in I_0} \lambda \|\widehat{f}_{mt} - f_{mt}^*\|_{\mathcal{H}_m}^2 \right). \tag{E.8}$$

**(Bound of the fourth term IV)** Using Equation (H.4) in the proof of Lemma (E.4) and $\lambda = \lambda_3$, we have

$$
\text{IV} \leq \sum_{m \in I_0} \frac{2\lambda_3^{\frac{1}{2}} \sqrt{\sum_{t=1}^T \|f_{mt}^*\|_{\mathcal{H}_m}^2}}{C_\alpha} \left( \sqrt{\sum_{t=1}^T \|\widehat{f}_{mt} - f_{mt}^*\|_{L_2(\Pi)}^2} + \lambda_3^{\frac{1}{2}} \sqrt{\sum_{t=1}^T \|\widehat{f}_{mt} - f_{mt}^*\|_{\mathcal{H}_m}^2} \right)
$$

$$
\leq \sum_{m \in I_0} \frac{(1 - \rho(I_0)^2))\kappa(I_0)}{16} \left( \sqrt{\sum_{t=1}^T \|\widehat{f}_{mt} - f_{mt}^*\|_{L_2(\Pi)}^2} + \lambda^{\frac{1}{2}} \sqrt{\sum_{t=1}^T \|\widehat{f}_{mt} - f_{mt}^*\|_{\mathcal{H}_m}^2} \right)^2
$$

$$
+ \frac{16\lambda_3 \sum_{m \in I_0} \sum_{t=1}^T \|f_{mt}^*\|_{\mathcal{H}_m}^2}{C_\alpha^2(1 - \rho(I_0)^2))\kappa(I_0)}.
$$

Thus, by Lemma A.1, we get the following bound

$$
\text{IV} \leq \frac{16\lambda_3 \sum_{m \in I_0} \sum_{t=1}^T \|f_{mt}^*\|_{\mathcal{H}_m}^2}{C_\alpha^2(1 - \rho(I_0)^2))\kappa(I_0)} + \frac{1}{8} \sum_{t=1}^T \left( \|\widehat{f}_t - f_t^*\|_{L_2(\Pi)}^2 + \sum_{m \in I_0} \lambda\|\widehat{f}_{mt} - f_{mt}^*\|_{\mathcal{H}_m}^2 \right). \quad \text{(E.9)}
$$

Now we are ready to combine all the bounds. Substituting the inequalities (E.6), (E.7), (E.8) and (E.9) into Equation (E.3), using the relation $\lambda = \lambda_3$ and $C_\alpha \leq 1$, we obtain

$$
\sum_{t=1}^T \|\widehat{f}_t - f_t^*\|_{L_2(\Pi)}^2 + \frac{\lambda_3}{C_\alpha} \sum_{m \in I_0} \sum_{t=1}^T \|\widehat{f}_{mt} - f_{mt}^*\|_{\mathcal{H}_m}^2
$$

$$
\leq \frac{32}{C_\alpha^2(1 - \rho(I_0)^2))\kappa(I_0)} \left( d\lambda_1^2 + \lambda_3 \sum_{m \in I_0} \sum_{t=1}^T \|f_{mt}^*\|_{\mathcal{H}_m}^2 \right) + \frac{9d\lambda_1^2}{C_\alpha^2(1 - \rho(I_0)^2))\kappa(I_0)}
$$

$$
+ \frac{16\lambda_3 \sum_{m \in I_0} \sum_{t=1}^T \|f_{mt}^*\|_{\mathcal{H}_m}^2}{C_\alpha^2(1 - \rho(I_0)^2))\kappa(I_0)} + \frac{1}{2} \sum_{t=1}^T \left( \|\widehat{f}_t - f_t^*\|_{L_2(\Pi)}^2 + \sum_{m \in I_0} \frac{\lambda_3}{C_\alpha}\|\widehat{f}_{mt} - f_{mt}^*\|_{\mathcal{H}_m}^2 \right).
$$

Moving the term $\frac{1}{2} \sum_{t=1}^T \left( \|\widehat{f}_t - f_t^*\|_{L_2(\Pi)}^2 + \sum_{m \in I_0} \frac{\lambda_3}{C_\alpha}\|\widehat{f}_{mt} - f_{mt}^*\|_{\mathcal{H}_m}^2 \right)$ to the LHS, we have

$$
\sum_{t=1}^T \|\widehat{f}_t - f_t^*\|_{L_2(\Pi)}^2 \leq \frac{96}{C_\alpha^2(1 - \rho(I_0)^2))\kappa(I_0)} \left( d\lambda_1^2 + \lambda_3 \sum_{m \in I_0} \sum_{t=1}^T \|f_{mt}^*\|_{\mathcal{H}_m}^2 \right),
$$

which is our assertion. □

By setting $\lambda$ in Theorem E.5, we get the convergence rate of the elastic-net model and the $L_1$ model. For every $1 \leq p < \infty$ we define the mixed$(2, p)$-norm of $f_{mt}$ as

$$
R_{2,p,f} = \left( \sum_{m=1}^M \left( \sum_{t=1}^T \|f_{mt}\|_{\mathcal{H}_m}^2 \right)^{\frac{p}{2}} \right)^{\frac{1}{p}}.
$$

**Corollary E.6.** Suppose all assumptions are satisfied and set $\lambda = \left( \frac{d}{n} \right)^{\frac{1}{1+s}} \left( \frac{R_{2,2,f^*}^2}{T} \right)^{-\frac{1}{1+s}}$. Set $\lambda_1, \lambda_2$ and $\lambda_3$ as $\lambda_1 = 4\sqrt{c_\nu \omega_n(T + \sqrt{T} \log M)u}, \lambda_2 = \lambda_1 \lambda^{\frac{1}{2}}, \lambda_3 = \lambda$ or $\lambda_3 = 0$. Under the condition that

$$
\widetilde{C}_1 c_s \sqrt{n} \xi_n(\lambda)^2 d \leq 1, \quad \text{(E.10)}
$$

there exist constants $\widetilde{C}_1, \widetilde{C}_2$ depending on $C_\alpha$, s, c, $C_1, \rho(I_0), \kappa(I_0)$ such that

$$
\sum_{t=1}^T \|\widehat{f}_t - f_t^*\|_{L_2(\Pi)}^2
$$

$$
\leq \widetilde{C}_2(T + \sqrt{T} \log M)\left( \frac{d}{n^{1/(1+s)}} + \left( \frac{d}{n} \right)^{\frac{1}{1+s}} \left( \frac{R_{2,2,f^*}^2}{T} \right)^{\frac{s}{1+s}} + \left( \frac{d^s}{n} \right)^{\frac{1}{1+s}} \left( \frac{R_{2,2,f^*}^2}{T} \right)^{\frac{1}{1+s}} \right) \quad \text{(E.11)}
$$

with probability at least $1 - \exp(-u) - \frac{M}{n^{s/(1+s)}} \sqrt{\exp\left(-(T^{1/2} \wedge n^{1/2}T^{1/4})\right)} - \exp\left( - \zeta_{nT}\left( \frac{1}{\widetilde{C}_1 d}, \lambda \right) \right)$ for all $u \geq 1$.

*Proof of Corollary E.6.* We start from the following relation, which is satisfied under our assumption shown later.

$$\frac{256 \max(c_s \sqrt{n} \xi_n^2, r)\Big(d + \frac{\lambda_3 \sum_{m \in I_0} \sum_{t=1}^{T} \|f_{mt}^*\|_{\mathcal{H}_m}^2}{\lambda_1^2}\Big)}{(1 - \rho(I_0)^2))\kappa(I_0)} \leq \frac{1}{8}. \tag{E.12}$$

Then we can apply Theorem E.5 since the assumptions are met. Theorem E.5 implies that

$$\sum_{t=1}^{T} \|\widehat{f}_t - f_t^*\|_{L_2(\Pi)}^2 \lesssim d\lambda_1^2 + \lambda_3 R_{2,2,f^*}^2.$$

Remind the definition of $\omega_n$. We have

$$\lambda_1^2 = 16 c_\nu \Big(\frac{1}{n} \vee \sqrt{\frac{6}{n^{2/(1+s)}} + \frac{C_1^4}{n^{(3+s)/(1+s)}\lambda^{2s}}} \vee \frac{\lambda^{-s}}{n} \vee \frac{\lambda^{-1}}{n^{\frac{2}{1+s}}}\Big)(T + \sqrt{T}\log M)u$$

$$\leq 16(C_1^2 \vee \sqrt{6})c_\nu\Big(\frac{1}{n^{1/(1+s)}} \vee \frac{\lambda^{-s}}{n} \vee \frac{\lambda^{-1}}{n^{\frac{2}{1+s}}}\Big)(T + \sqrt{T}\log M)u.$$

When $\lambda = \Big(\frac{d}{n}\Big)^{\frac{1}{1+s}}\Big(\frac{R_{2,2,f^*}^2}{T}\Big)^{-\frac{1}{1+s}}$, we get

$$d\lambda_1^2 \leq 16(C_1^2 \vee \sqrt{6})c_\nu d\Big(\frac{1}{n^{1/(1+s)}} \vee \frac{\lambda^{-s}}{n} \vee \frac{\lambda^{-1}}{n^{\frac{2}{1+s}}}\Big)(T + \sqrt{T}\log M)u$$

$$= 16(C_1^2 \vee \sqrt{6})c_\nu(T + \sqrt{T}\log M)u$$

$$\cdot \Big(\frac{d}{n^{1/(1+s)}} \vee \Big(\frac{d}{n}\Big)^{\frac{1}{1+s}}\Big(\frac{R_{2,2,f^*}^2}{T}\Big)^{\frac{s}{1+s}} \vee \Big(\frac{d^s}{n}\Big)^{\frac{1}{1+s}}\Big(\frac{R_{2,2,f^*}^2}{T}\Big)^{\frac{1}{1+s}}\Big),$$

and

$$\lambda_3 R_{2,2,f^*}^2 = \frac{1}{T}\Big(\frac{d}{n}\Big)^{\frac{1}{1+s}}\Big(\frac{R_{2,2,f^*}^2}{T}\Big)^{\frac{s}{1+s}}.$$

Therefore we have

$$\sum_{t=1}^{T} \|\widehat{f}_t - f_t^*\|_{L_2(\Pi)}^2 \leq \frac{96}{C_\alpha^2(1 - \rho(I_0)^2))\kappa(I_0)}\Big(d\lambda_1^2 + \lambda_3 \sum_{m \in I_0} \sum_{t=1}^{T} \|f_{mt}^*\|_{\mathcal{H}_m}^2\Big)$$

$$\leq \frac{96(16(C_1^2 \vee \sqrt{6})c_\nu + 1)u}{C_\alpha^2(1 - \rho(I_0)^2))\kappa(I_0)}(T + \sqrt{T}\log M)$$

$$\cdot \Big(\frac{d}{n^{1/(1+s)}} \vee \Big(\frac{d}{n}\Big)^{\frac{1}{1+s}}\Big(\frac{R_{2,2,f^*}^2}{T}\Big)^{\frac{s}{1+s}} \vee \Big(\frac{d^s}{n}\Big)^{\frac{1}{1+s}}\Big(\frac{R_{2,2,f^*}^2}{T}\Big)^{\frac{1}{1+s}}\Big).$$

Thus by setting $\widetilde{C}_2$ as

$$\widetilde{C}_2 = \frac{96(16(C_1^2 \vee \sqrt{6})c_\nu + 1)u}{C_\alpha^2(1 - \rho(I_0)^2))\kappa(I_0)},$$

we obtain the inequality (E.11). At last, we show the condition (E.10) yields the condition (E.12) for some $\widetilde{C}_1$ and $r$. Note that

$$\frac{\lambda_3 \sum_{m \in I_0} \sum_{t=1}^{T} \|f_{mt}^*\|_{\mathcal{H}_m}^2}{\lambda_1^2} \leq d^{\frac{1}{1+s}} n^{-\frac{1}{1+s}} R_{2,2,f^*}^{\frac{2s}{1+s}} T^{\frac{1}{1+s}} / c_\nu d^{-\frac{s}{1+s}} n^{-\frac{1}{1+s}} R_{2,2,f^*}^{\frac{2s}{1+s}} T^{\frac{1}{1+s}} u$$

$$\leq \frac{d}{c_\nu u} \leq d,$$

since $c_\nu \geq 1, u \geq 1$ by definition. Therefore the condition (E.12) holds if

$$\frac{256 \max(c_s \sqrt{n} \xi_n^2, r)(d + d)}{(1 - \rho(I_0)^2))\kappa(I_0)} \leq \frac{1}{8}$$

holds. Thus by setting $\widetilde{C}_1 = \frac{4096}{(1-\rho(I_0)^2))\kappa(I_0)}$ and $r = \frac{1}{\widetilde{C}_1 d}$, the condition (E.10) yields the condition (E.12). □

With further simplification, we derive the main result presented.

*Proof of Theorem 6.1.* Note that under the Union Bound Assumption and $\|f_{mt}^*\|_{\mathcal{H}_m} \geq c$ for some constant $c > 0$ for all $m$ and $t$, we have

$$c^2 d \leq T^{-1} R_{2,2,f^*}^2 \leq C^2 d.$$

Thus, $\lambda = d^{\frac{1}{1+s}} n^{-\frac{1}{1+s}} T^{\frac{1}{1+s}} R_{2,2,f^*}^{-\frac{2}{1+s}} \asymp n^{-\frac{1}{1+s}}$, which leads to

$$\omega_n = \max\left(\frac{1}{n}, \sqrt{\frac{6}{n^{2/(1+s)}} + \frac{C_1^4}{n^{(3+s)/(1+s)}\lambda^{2s}}}, \frac{\lambda^{-s}}{n}, \frac{\lambda^{-1}}{n^{\frac{2}{1+s}}}\right) \asymp n^{-\frac{1}{1+s}}.$$

We derive the setting of parameters by substituting these results into Corollary E.6. Similarly, we have

$$\frac{d}{n^{1/(1+s)}} + \left(\frac{d}{n}\right)^{\frac{1}{1+s}} \left(\frac{R_{2,2,f^*}^2}{T}\right)^{\frac{s}{1+s}} + \left(\frac{d^s}{n}\right)^{\frac{1}{1+s}} \left(\frac{R_{2,2,f^*}^2}{T}\right)^{\frac{1}{1+s}} \leq C \frac{d}{n^{1/(1+s)}}$$

for some constant $C > 0$. Note that $\frac{d \log M}{\sqrt{n}} + \frac{d}{n^{(1-s)/2(1+s)}} = o(1)$ implies $\widetilde{C}_1 c_s \sqrt{n} \xi_n(\lambda)^2 d \leq 1$, so the conditions in Corollary E.6 are satisfied. Thus, we complete the proof of the theorem. $\qquad\square$

## F    PROOF OF LEMMA A.1

*Proof of Lemma A.1.* For $t = 1, \ldots, T$, we have

$$\begin{aligned}
\|f_t\|_{L_2(\Pi)}^2 &= \|f_{I_t}\|_{L_2(\Pi)}^2 + 2\langle f_{I_t}, f_{I_t^c}\rangle_{L_2(\Pi)} + \|f_{I_t^c}\|_{L_2(\Pi)}^2 \\
&\geq \|f_{I_t}\|_{L_2(\Pi)}^2 - 2\rho(I_t)\|f_{I_t}\|_{L_2(\Pi)}\|f_{I_t^c}\|_{L_2(\Pi)} + \|f_{I_t^c}\|_{L_2(\Pi)}^2 \\
&\geq (1 - \rho(I_t)^2)\|f_{I_t}\|_{L_2(\Pi)}^2 \geq (1 - \rho(I_t)^2)\kappa(I_t) \sum_{m \in I_t} \|f_{mt}\|_{L_2(\Pi)}^2.
\end{aligned}$$

Thus, we get the assertion by summation using the definition of $\kappa(I_0)$ and $\rho(I_0)$. $\qquad\square$

## G    PROOF OF THEOREM E.2 AND THEOREM E.3

First, we show the proof of event $\mathcal{E}_1(u)$. We start by introducing the following proposition given in Suzuki & Sugiyama [65].

**Proposition G.1.** Under the Bounded Kernel Assumption, the Spectral Assumption and the Sup-norm Assumption, we have for all $\lambda > 0$ and all $u \geq 1$

$$P\left(\sup_{f_{mt} \in \mathcal{H}_m} \frac{\left|\frac{1}{n}\sum_{i=1}^n \epsilon_{ti} f_{mt}(\boldsymbol{x}_{ti})\right|}{\|f_{mt}\|_{L_2(\Pi)} + \lambda^{\frac{1}{2}}\|f_{mt}\|_{\mathcal{H}_m}} \geq K\left[2C_s L\Xi_n + \sqrt{\frac{L^2 h}{n}} + \frac{C_1 L \lambda^{-\frac{s}{2}} h}{n}\right]\right) \leq e^{-h},$$

where $|\epsilon_{ti}| \leq L$, $\Xi_n = \max\left(\frac{\lambda^{-\frac{s}{2}}}{\sqrt{n}}, \frac{\lambda^{-\frac{1}{2}}}{n^{\frac{1}{1+s}}}, \sqrt{\frac{1}{n}}\right)$, $C_s$ is a constant depending on $s$ and $K$ is the constant appeared in Talagrand's concentration inequality.

We define

$$c_s := \max\left(2K\left(C_s + 1 + C_1\right), K\left[8K\left(C_s + 1 + C_1\right) + C_1 + C_1^2\right], 1\right).$$

Note that $|\epsilon_{ti}| \leq 1$ by definition, we have

$$K\left[2C_s L\Xi_n + \sqrt{\frac{L^2 h}{n}} + \frac{C_1 L \lambda^{-\frac{s}{2}} h}{n}\right] \leq K\left(2C_s + \sqrt{h} + C_1 \frac{h}{\sqrt{n}}\right)\Xi_n \leq c_s \Xi_n \eta(h),$$

where $\eta(h) := \max(1, \sqrt{h}, \frac{h}{\sqrt{n}})$.

Now we are ready for the proof of event $\mathcal{E}_1(u)$. Using the truncation method, we obtain the probability bound of Theorem E.2.

*Proof of Theorem E.2.* Using the Cauchy-Schwarz inequality, we have

$$
\frac{1}{n}\sum_{t=1}^{T}\sum_{i=1}^{n}\epsilon_{ti}f_{mt}(\boldsymbol{x}_{ti}) = \frac{1}{n}\sum_{t=1}^{T}\sum_{i=1}^{n}\epsilon_{ti}\frac{f_{mt}(\boldsymbol{x}_{ti})}{\|f_{mt}\|}\|f_{mt}\|
$$

$$
\leq \sqrt{\sum_{t=1}^{T}\left(\frac{1}{n}\sum_{i=1}^{n}\epsilon_{ti}\frac{f_{mt}(\boldsymbol{x}_{ti})}{\|f_{mt}\|}\right)^2}\sqrt{\sum_{t=1}^{T}\|f_{mt}\|^2}.
$$

Set $\|f_{mt}\| \leftarrow \|f_{mt}\|_{L_2(\Pi)} + \lambda^{\frac{1}{2}}\|f_{mt}\|_{\mathcal{H}_m}$. By $(x+y)^2 \leq 2(x^2+y^2)$, we have

$$
\sqrt{\sum_{t=1}^{T}\|f_{mt}\|^2} = \sqrt{\sum_{t=1}^{T}(\|f_{mt}\|_{L_2(\Pi)} + \lambda^{\frac{1}{2}}\|f_{mt}\|_{\mathcal{H}_m})^2}
$$

$$
\leq \sqrt{2}\left(\sqrt{\sum_{t=1}^{T}\|f_{mt}\|^2_{L_2(\Pi)}} + \lambda^{\frac{1}{2}}\sqrt{\sum_{t=1}^{T}\|f_{mt}\|^2_{\mathcal{H}_m}}\right),
$$

which generates the desired norm part appeared in the RHS of event $\mathcal{E}_1(u)$. Notice that now

$$
\frac{1}{n}\sum_{i=1}^{n}\epsilon_{ti}\frac{f_{mt}(\boldsymbol{x}_{ti})}{\|f_{mt}\|} = \frac{1}{n}\sum_{i=1}^{n}\frac{\epsilon_{ti}f_{mt}(\boldsymbol{x}_{ti})}{\|f_{mt}\|_{L_2(\Pi)} + \lambda^{\frac{1}{2}}\|f_{mt}\|_{\mathcal{H}_m}}.
$$

We will use the following notation for further discussion:

$$
X_{mt} := \sup_{f_{mt}\in\mathcal{H}_m}\frac{\frac{1}{n}\sum_{i=1}^{n}\epsilon_{ti}f_{mt}(\boldsymbol{x}_{ti})}{\|f_{mt}\|_{L_2(\Pi)} + \lambda^{\frac{1}{2}}\|f_{mt}\|_{\mathcal{H}_m}},
$$

where $X_{mt}$ is a sub-exponential random variable. Notice that $|\epsilon_{ti}| \leq 1$ by definition. With the Bounded Kernel Assumption, we have

$$
\mathbb{E}\Big[X_{mt}\Big] = \mathbb{E}\Big[\sup_{f_{mt}\in\mathcal{H}_m}\frac{\frac{1}{n}\sum_{i=1}^{n}\epsilon_{ti}f_{mt}(\boldsymbol{x}_{ti})}{\|f_{mt}\|_{L_2(\Pi)} + \lambda^{\frac{1}{2}}\|f_{mt}\|_{\mathcal{H}_m}}\Big] = 0,
$$

$$
\mathbb{E}\Big[X_{mt}^2\Big] \leq \mathbb{E}\Big[\sup_{f_{mt}\in\mathcal{H}_m}\frac{\frac{1}{n^2}\sum_{i=1}^{n}\epsilon_{ti}^2 f_{mt}(\boldsymbol{x}_{ti})^2}{\|f_{mt}\|^2_{L_2(\Pi)} + \lambda\|f_{mt}\|^2_{\mathcal{H}_m}}\Big] \leq \frac{1}{n},
$$

$$
\mathbb{E}\Big[X_{mt}^4\Big] = \mathbb{E}\Big[\sup_{f_{mt}\in\mathcal{H}_m}\frac{\frac{1}{n^4}(\sum_{i=1}^{n}\epsilon_{ti}^4 f_{mt}(\boldsymbol{x}_{ti})^4 + 6\sum_{i=1}^{n}\sum_{j\neq i}^{n}\epsilon_{ti}^2 f_{mt}(\boldsymbol{x}_{ti})^2\epsilon_{tj}^2 f_{mt}(\boldsymbol{x}_{tj})^2)}{(\|f_{mt}\|_{L_2(\Pi)} + \lambda^{\frac{1}{2}}\|f_{mt}\|_{\mathcal{H}_m})^4}\Big] \leq \frac{6}{n^2} + \frac{C_1^4}{n^3\lambda^{2s}},
$$

where we use $\|f_{mt}\|_{\infty} \leq C_1\lambda^{-\frac{s}{2}}(\|f_{mt}\|_{L_2(\Pi)} + \lambda^{\frac{1}{2}}\|f_{mt}\|_{\mathcal{H}_m})$ in the last line, a direct result from the Sup-norm Assumption and Young's inequality. Using the truncation method, we have

$$
\mathbb{P}\Big(\sum_{t=1}^{T}X_{mt}^2 - \mathbb{E}\Big[\sum_{t=1}^{T}X_{mt}^2\Big] \geq 2u\Big) \leq \mathbb{P}\Big(\sum_{t=1}^{T}X_{mt}^2\mathbb{I}\{|X_{mt}| \leq B\} - \mathbb{E}\Big[\sum_{t=1}^{T}X_{mt}^2\mathbb{I}\{|X_{mt}| \leq B\}\Big] \geq u\Big)
$$

$$
+ \mathbb{P}\Big(\sum_{t=1}^{T}X_{mt}^2\mathbb{I}\{|X_{mt}| \geq B\} - \mathbb{E}\Big[\sum_{t=1}^{T}X_{mt}^2\mathbb{I}\{|X_{mt}| \geq B\}\Big] \geq u\Big),
$$

for all $B \geq 0$. For simplicity, we use the following notations:

$$
Y_{mt} := X_{mt}^2\mathbb{I}\{|X_{mt}| \geq B\} - \mathbb{E}\Big[X_{mt}^2\mathbb{I}\{|X_{mt}| \geq B\}\Big],
$$

$$
Z_{mt} := X_{mt}^2\mathbb{I}\{|X_{mt}| \leq B\} - \mathbb{E}\Big[X_{mt}^2\mathbb{I}\{|X_{mt}| \leq B\}\Big],
$$

$$
\psi := \frac{6}{n^2} + \frac{C_1^4}{n^3\lambda^{2s}}.
$$

We will develop the probability bounds for $Y_{mt}$ and $Z_{mt}$ separately with Markov's inequality and Bernstein's concentration inequality. Using Markov's inequality, we have

$$\mathbb{P}\Big(\Big|\sum_{t=1}^{T} Y_{mt}\Big| \geq u\Big) \leq \frac{\mathbb{E}\Big[\Big|\sum_{t=1}^{T} Y_{mt}\Big|\Big]}{u}$$

$$\leq \frac{2T}{u}\mathbb{E}\Big[X_{mt}^2\mathbb{I}\{|X_{mt}| \geq B\}\Big].$$

We apply the Cauchy-Schwarz inequality to the RHS of the above inequality and obtain

$$\mathbb{P}\Big(\Big|\sum_{t=1}^{T} Y_{mt}\Big| \geq u\Big) \leq \frac{2T}{u}\sqrt{\mathbb{E}[X_{mt}^4]\mathbb{P}(|X_{mt}| \geq B)}$$

$$\leq \frac{2T}{u}\sqrt{\psi\mathbb{P}(|X_{mt}| \geq B)}.$$

Using Proposition G.1, we have

$$\mathbb{P}(|X_{mt}| \geq c_s\Xi_n\eta(h)) \leq \exp(-h)$$

by setting $B \leftarrow c_s\Xi_n\eta(h)$. Therefore, with probability at least $1 - k$, we have

$$\Big|\sum_{t=1}^{T} Y_{mt}\Big| \leq \frac{2T\sqrt{\psi\exp(-h)}}{k}.$$

As for $Z_{mt}$, we use the following version of Bernstein's concentration inequality [4].

**Proposition G.2.** Let $X_1, \ldots, X_n$ be iid and suppose that $|X_i| \leq c$, $\mathbb{E}(X_i) = \mu$ and $\mathrm{Var}(X_i) = \sigma^2$. With probability at least $1 - \delta$,

$$\Big|\bar{X}_n - \mu\Big| \leq \sqrt{\frac{2\sigma^2\log(1/\delta)}{n}} + \frac{2c\log(1/\delta)}{3n}.$$

Directly applying the above proposition to $Z_{mt}$, we obtain with probability at least $1 - \exp(-u)$,

$$\Big|\sum_{t=1}^{T} Z_{mt}\Big| \leq \sqrt{2\psi Tu} + \frac{2}{3}(c_s\Xi_n\eta(h))^2 u.$$

Combining both parts and use the fact that $\mathbb{E}\Big[\sum_{t=1}^{T} X_{mt}^2\Big] = T\mathbb{E}\Big[X_{mt}^2\Big] \leq T/n$, we get

$$\sum_{t=1}^{T} X_{mt}^2 \leq \frac{T}{n} + \sqrt{2\psi Tu} + \frac{2}{3}(c_s\Xi_n\eta(h))^2 u + \frac{2T\sqrt{\psi\exp(-h)}}{k}$$

with probability at least $1 - \exp(-u) - k$. Therefore, the uniform bound of all $m = 1, \ldots, M$ is given as

$$\max_m \sum_{t=1}^{T} X_{mt}^2 \leq \frac{T}{n} + \sqrt{2\psi Tu} + \frac{2}{3}(c_s\Xi_n\eta(h))^2 u + \frac{2T\sqrt{\psi\exp(-h)}}{k}$$

with probability at least $1 - M(\exp(-u) + k)$. If we set $u \leftarrow u + \log M$ and $k \leftarrow \frac{k}{M}$, then with probability at least $1 - \exp(-u) - k$,

$$\max_m \sum_{t=1}^{T} X_{mt}^2 \leq \frac{T}{n} + \sqrt{2\psi T(u + \log M)} + \frac{2}{3}(c_s\Xi_n\eta(h))^2(u + \log M) + \frac{2MT\sqrt{\psi\exp(-h)}}{k}.$$

Here we fix $k$ as $k \leftarrow \frac{M}{n^{s/(1+s)}}\sqrt{\exp(-h)}$, so with probability at least $1 - \exp(-u) - \frac{M}{n^{s/(1+s)}}\sqrt{\exp(-h)}$,

$$\max_m \sum_{t=1}^{T} X_{mt}^2 \leq \frac{T}{n} + \sqrt{2\psi T(u + \log M)} + \frac{2}{3}(c_s\Xi_n\eta(h))^2(u + \log M) + 2Tn^{\frac{s}{1+s}}\sqrt{\psi}.$$

Recall that $\omega_n = \max\left(\frac{1}{n}, \sqrt{\frac{6}{n^{2/(1+s)}} + \frac{C_1^4}{n^{(3+s)/(1+s)}\lambda^{2s}}}, \frac{\lambda^{-s}}{n}, \frac{\lambda^{-1}}{n^{\frac{2}{1+s}}}\right) = n^{\frac{s}{1+s}}\sqrt{\psi} \vee \Xi_n^2$. Through basic calculation we obtain

$$\max_m \sum_{t=1}^{T} X_{mt}^2 \le \omega_n\left(3T + \sqrt{2Tu} + \frac{\sqrt{2T\log M}}{n^{s/(1+s)}}\right) + \frac{2}{3}c_s^2\omega_n\eta(h)^2(u + \log M).$$

Let $h \leftarrow T^{1/2} \wedge n^{1/2}T^{1/4}$, then we have $\eta(h)^2 = (1 \vee \sqrt{h} \vee \frac{h}{\sqrt{n}})^2 = 1 \vee \sqrt{T}$, which gives

$$\max_m \sum_{t=1}^{T} X_{mt}^2 \le (3 + \sqrt{2} + \frac{2}{3}c_s^2)\omega_n T(1 \vee u) + (\frac{2}{3}c_s^2 + \sqrt{2})\omega_n \log M \sqrt{T},$$

where we use $\sqrt{u} \le 1 \vee u, \sqrt{\log M} \le \log M, 1 \le \sqrt{T} \le T$. We define $c_\nu := 2(3 + \sqrt{2} + \frac{2}{3}c_s^2)$. Thus, the above inequality can be further simplified as

$$\max_m \sum_{t=1}^{T} X_{mt}^2 \le \frac{c_\nu}{2}\omega_n(T + \sqrt{T}\log M)(1 \vee u).$$

Therefore, for all $u \ge 1$, with probability at least $1 - \frac{M}{n^{s/(1+s)}}\sqrt{\exp\left(-(T^{1/2} \wedge n^{1/2}T^{1/4})\right)} - \exp(-u)$,

$$\max_m \left|\frac{1}{n}\sum_{t=1}^{T}\sum_{i=1}^{n}\epsilon_{ti}f_{mt}(\boldsymbol{x}_{ti})\right| \le \max_m \sqrt{2\sum_{t=1}^{T}X_{mt}^2}\left(\sqrt{\sum_{t=1}^{T}\|f_{mt}\|_{L_2(\Pi)}^2} + \lambda^{\frac{1}{2}}\sqrt{\sum_{t=1}^{T}\|f_{mt}\|_{\mathcal{H}_m}^2}\right)$$

$$\le \sqrt{c_\nu\omega_n(T + \sqrt{T}\log M)u}\left(\sqrt{\sum_{t=1}^{T}\|f_{mt}\|_{L_2(\Pi)}^2} + \lambda^{\frac{1}{2}}\sqrt{\sum_{t=1}^{T}\|f_{mt}\|_{\mathcal{H}_m}^2}\right),$$

which is our assertion. □

Now we give the proof to our event $\mathcal{E}_2(r)$, which is indeed a direct extension to the following proposition [65].

**Proposition G.3.** Define event $\mathcal{E}(r)$ as

$$\mathcal{E}(r) = \left\{\left|\|\sum_{m=1}^{M}f_m\|_n^2 - \|\sum_{m=1}^{M}f_m\|_{L_2(\Pi)}^2\right| \le \max(c_s\sqrt{n}\xi_n^2, r)\left[\sum_{m=1}^{M}(\|f_m\|_{L_2(\Pi)} + \lambda^{\frac{1}{2}}\|f_m\|_{\mathcal{H}_m})\right]^2,\right.$$

$$\left.\forall f_m \in \mathcal{H}_m, \forall m = 1, \ldots, M\right\}.$$

Under the Spectral Assumption and the Sup-norm Assumption, we have for all $\lambda > 0$ and all $r \ge 1$

$$P(\mathcal{E}(r)) \ge 1 - \exp(-\zeta_n(r, \lambda)),$$

where $\zeta_n(r, \lambda) = \min\left(\frac{r^2\log M}{n\xi_n(\lambda)^4c_s^2}, \frac{r}{\xi_n(\lambda)^2c_s}\right)$.

The following lemma is a useful inequality used in the proof of event $\mathcal{E}_2(r)$.

**Lemma G.4.** For all $t = 1, \ldots, T$, and $m = 1, \ldots, M$, set $\lambda \ge 0$, we have the following inequality

$$\sum_{t=1}^{T}\left[\sum_{m=1}^{M}\left(\|f_{mt}\|_{L_2(\Pi)} + \lambda^{\frac{1}{2}}\|f_{mt}\|_{\mathcal{H}_m}\right)\right]^2 \le \left[\sum_{m=1}^{M}\left(\sqrt{\sum_{t=1}^{T}\|f_{mt}\|_{L_2(\Pi)}^2} + \lambda^{\frac{1}{2}}\sqrt{\sum_{t=1}^{T}\|f_{mt}\|_{\mathcal{H}_m}^2}\right)\right]^2.$$

(G.1)

*Proof of Lemma G.4.* Define $\boldsymbol{f}_m^{L_2(\Pi)} := \left[\|f_{m1}\|_{L_2(\Pi)}, \|f_{m2}\|_{L_2(\Pi)}, \dots, \|f_{mT}\|_{L_2(\Pi)}\right]$, and $\boldsymbol{f}_m^{\mathcal{H}_m}$ $:= \left[\|f_{m1}\|_{\mathcal{H}_m}, \|f_{m2}\|_{\mathcal{H}_m}, \dots, \|f_{mT}\|_{\mathcal{H}_m}\right]$. Then we have

$$
\left\| \sum_{m=1}^M \boldsymbol{f}_m^{L_2(\Pi)} + \lambda^{\frac{1}{2}} \sum_{m=1}^M \boldsymbol{f}_m^{\mathcal{H}_m} \right\|_2^2
$$

$$
= \left\| \left[ \sum_{m=1}^M \|f_{m1}\|_{L_2(\Pi)} + \lambda^{\frac{1}{2}} \sum_{m=1}^M \|f_{m1}\|_{\mathcal{H}_m}, \dots, \sum_{m=1}^M \|f_{mT}\|_{L_2(\Pi)} + \lambda^{\frac{1}{2}} \sum_{m=1}^M \|f_{mT}\|_{\mathcal{H}_m} \right] \right\|_2^2
$$

$$
= \sum_{t=1}^T \left[ \sum_{m=1}^M \left( \|f_{mt}\|_{L_2(\Pi)} + \lambda^{\frac{1}{2}} \|f_{mt}\|_{\mathcal{H}_m} \right) \right]^2,
$$

which is the LHS of the above inequality. Similarly, we have

$$
\left( \sum_{m=1}^M \|\boldsymbol{f}_m^{L_2(\Pi)}\|_2 + \lambda^{\frac{1}{2}} \sum_{m=1}^M \|\boldsymbol{f}_m^{\mathcal{H}_m}\|_2 \right)^2 = \left[ \sum_{m=1}^M \left( \sqrt{\sum_{t=1}^T \|f_{mt}\|_{L_2(\Pi)}^2} + \lambda^{\frac{1}{2}} \sqrt{\sum_{t=1}^T \|f_{mt}\|_{\mathcal{H}_m}^2} \right) \right]^2,
$$

which is the RHS. By the triangle inequality, we have

$$
\left\| \sum_{m=1}^M \boldsymbol{f}_m^{L_2(\Pi)} + \lambda^{\frac{1}{2}} \sum_{m=1}^M \boldsymbol{f}_m^{\mathcal{H}_m} \right\|_2 \leq \sum_{m=1}^M \|\boldsymbol{f}_m^{L_2(\Pi)}\|_2 + \lambda^{\frac{1}{2}} \sum_{m=1}^M \|\boldsymbol{f}_m^{\mathcal{H}_m}\|_2,
$$

which gives the assertion. □

Now we show the probability bound for event $\mathcal{E}_2(r)$.

*Proof of Theorem E.3.* Using Lemma G.1 and $\frac{\sum_{t=1}^T x_t}{\sum_{t=1}^T y_t} \leq \max_t \frac{x_t}{y_t}$, we have

$$
\sup_{f_{mt} \in \mathcal{H}_m} \frac{\sum_{t=1}^T \left| \left\| \sum_{m=1}^M f_{mt} \right\|_n^2 - \left\| \sum_{m=1}^M f_{mt} \right\|_{L_2(\Pi)}^2 \right|}{\left[ \sum_{m=1}^M \left( \sqrt{\sum_{t=1}^T \|f_{mt}\|_{L_2(\Pi)}^2} + \lambda^{\frac{1}{2}} \sqrt{\sum_{t=1}^T \|f_{mt}\|_{\mathcal{H}_m}^2} \right) \right]^2}
$$

$$
\leq \sup_{f_{mt} \in \mathcal{H}_m} \frac{\sum_{t=1}^T \left| \left\| \sum_{m=1}^M f_{mt} \right\|_n^2 - \left\| \sum_{m=1}^M f_{mt} \right\|_{L_2(\Pi)}^2 \right|}{\sum_{t=1}^T \left( \sum_{m=1}^M \left( \|f_{mt}\|_{L_2(\Pi)} + \lambda^{\frac{1}{2}} \|f_{mt}\|_{\mathcal{H}_m} \right) \right)^2}
$$

$$
\leq \sup_{f_{mt} \in \mathcal{H}_m} \max_t \frac{\left| \left\| \sum_{m=1}^M f_{mt} \right\|_n^2 - \left\| \sum_{m=1}^M f_{mt} \right\|_{L_2(\Pi)}^2 \right|}{\left( \sum_{m=1}^M \left( \|f_{mt}\|_{L_2(\Pi)} + \lambda^{\frac{1}{2}} \|f_{mt}\|_{\mathcal{H}_m} \right) \right)^2}.
$$

By Proposition G.3, we get

$$
\sup_{f_{mt} \in \mathcal{H}_m} \max_t \frac{\left| \left\| \sum_{m=1}^M f_{mt} \right\|_n^2 - \left\| \sum_{m=1}^M f_{mt} \right\|_{L_2(\Pi)}^2 \right|}{\left( \sum_{m=1}^M \left( \|f_{mt}\|_{L_2(\Pi)} + \lambda^{\frac{1}{2}} \|f_{mt}\|_{\mathcal{H}_m} \right) \right)^2} \leq \max(c_s \sqrt{n} \xi_n^2, r)
$$

with probability at least $1 - T \exp(-\zeta_n(r, \lambda))$. Substituting $T \exp(-\zeta_n(r, \lambda))$ with $\zeta_{nT}$, we get the desired result. □

# H    PROOF OF LEMMA E.4

*Proof of Lemma E.4.* On the event $\mathcal{E}_2(r)$, for all $f_{mt} \in \mathcal{H}_m$, we have

$$
\sum_{t=1}^T |\|f_{mt}\|_n^2 - \|f_{mt}\|_{L_2(\Pi)}^2| \leq \sum_{t=1}^T \max(c_s \sqrt{n} \xi_n^2, r)(\|f_{mt}\|_{L_2(\Pi)} + \lambda^{\frac{1}{2}} \|f_{mt}\|_{\mathcal{H}_m})^2.
$$

Therefore, we can obtain the upper bound of the regularization term as

$$\lambda_1 \sum_{m=1}^{M} \sqrt{\sum_{t=1}^{T} \|f_{mt}\|_n^2} \le \lambda_1 \sum_{m=1}^{M} \sqrt{\sum_{t=1}^{T} \left[ \|f_{mt}\|_{L_2(\Pi)}^2 + \max(c_s\sqrt{n}\xi_n^2, r)(\|f_{mt}\|_{L_2(\Pi)} + \lambda^{\frac{1}{2}}\|f_{mt}\|_{\mathcal{H}_m})^2 \right]}$$

$$\le \lambda_1 \sum_{m=1}^{M} \sqrt{\sum_{t=1}^{T} \left[ \|f_{mt}\|_{L_2(\Pi)}^2 + 2\max(c_s\sqrt{n}\xi_n^2, r)(\|f_{mt}\|_{L_2(\Pi)}^2 + \lambda\|f_{mt}\|_{\mathcal{H}_m}^2) \right]}$$

$$\le \lambda_1 \sum_{m=1}^{M} \sqrt{\frac{5}{4}\sum_{t=1}^{T} \|f_{mt}\|_{L_2(\Pi)}^2 + \frac{\lambda}{4}\sum_{t=1}^{T} \|f_{mt}\|_{\mathcal{H}_m}^2},$$

since $\max(\phi\sqrt{n}\xi_n^2, r) \le \frac{1}{8}$. Combined with the regularization term $\lambda_2$, we have

$$\lambda_1 \sum_{m=1}^{M} \sqrt{\sum_{t=1}^{T} \|f_{mt}\|_n^2} + \lambda_2 \sum_{m=1}^{M} \sqrt{\sum_{t=1}^{T} \|f_{mt}\|_{\mathcal{H}_m}^2} \le \frac{3}{2}\left( \lambda_1 \sum_{m=1}^{M} \sqrt{\sum_{t=1}^{T} \|f_{mt}\|_{L_2(\Pi)}^2} + \lambda_2 \sum_{m=1}^{M} \sqrt{\sum_{t=1}^{T} \|f_{mt}\|_{\mathcal{H}_m}^2} \right),$$

(H.1)

because $\lambda_2 = \lambda^{\frac{1}{2}}\lambda_1$. Similarly, we can obtain the lower bound of the regularization term as

$$\lambda_1 \sum_{m=1}^{M} \sqrt{\sum_{t=1}^{T} \|f_{mt}\|_n^2} \ge \lambda_1 \sum_{m=1}^{M} \sqrt{\sum_{t=1}^{T} \max\left( (\|f_{mt}\|_{L_2(\Pi)}^2 - 2\max(c_s\sqrt{n}\xi_n^2, r))(\|f_{mt}\|_{L_2(\Pi)}^2 + \lambda\|f_{mt}\|_{\mathcal{H}_m}^2), 0 \right)}$$

$$\ge \lambda_1 \sum_{m=1}^{M} \sqrt{\sum_{t=1}^{T} \max\left( \frac{3}{4}\|f_{mt}\|_{L_2(\Pi)}^2 - \frac{\lambda}{4}\|f_{mt}\|_{\mathcal{H}_m}^2, 0 \right)}$$

$$\ge \frac{1}{2}\lambda_1 \sum_{m=1}^{M} \sqrt{\sum_{t=1}^{T} \|f_{mt}\|_{L_2(\Pi)}^2} - \frac{\lambda_1\lambda^{\frac{1}{2}}}{2} \sum_{m=1}^{M} \sqrt{\sum_{t=1}^{T} \|f_{mt}\|_{\mathcal{H}_m}^2},$$

where in the second inequality we use the relation $\sqrt{\max(\frac{3}{4}x^2 - \frac{1}{4}y^2, 0)} \ge \frac{x-y}{2}$ for all $x, y \ge 0$. Combined with the regularization term $\lambda_2$, we have

$$\lambda_1 \sum_{m=1}^{M} \sqrt{\sum_{t=1}^{T} \|f_{mt}\|_n^2} + \lambda_2 \sum_{m=1}^{M} \sqrt{\sum_{t=1}^{T} \|f_{mt}\|_{\mathcal{H}_m}^2}$$

$$\ge \frac{1}{2}\left( \lambda_1 \sum_{m=1}^{M} \sqrt{\sum_{t=1}^{T} \|f_{mt}\|_{L_2(\Pi)}^2} + \lambda_2 \sum_{m=1}^{M} \sqrt{\sum_{t=1}^{T} \|f_{mt}\|_{\mathcal{H}_m}^2} \right).$$

(H.2)

Continuing from Equation (E.1), we have

$$\frac{1}{n}\sum_{t=1}^{T}\sum_{i=1}^{n} \epsilon_{ti}(\widehat{f}_t(\boldsymbol{x}_{ti}) - f_t^*(\boldsymbol{x}_{ti})) + \frac{\exp(-C)}{2(1 + \exp(C))^2}\frac{1}{n}\sum_{t=1}^{T}\sum_{i=1}^{n} (\widehat{f}_t(\boldsymbol{x}_{ti}) - f_t^*(\boldsymbol{x}_{ti}))^2$$

$$+ \lambda_1 \sum_{m=1}^{M} \sqrt{\sum_{t=1}^{T} \|\widehat{f}_{mt}\|_n^2} + \lambda_2 \sum_{m=1}^{M} \sqrt{\sum_{t=1}^{T} \|\widehat{f}_{mt}\|_{\mathcal{H}_m}^2} + \lambda_3 \sum_{m=1}^{M}\sum_{t=1}^{T} \|\widehat{f}_{mt}\|_{\mathcal{H}_m}^2$$

$$\le \lambda_1 \sum_{m \in I_0} \sqrt{\sum_{t=1}^{T} \|f_{mt}^*\|_n^2} + \lambda_2 \sum_{m \in I_0} \sqrt{\sum_{t=1}^{T} \|f_{mt}^*\|_{\mathcal{H}_m}^2} + \lambda_3 \sum_{m \in I_0}\sum_{t=1}^{T} \|f_{mt}^*\|_{\mathcal{H}_m}^2.$$

Remind that $C_\alpha = \frac{\exp(-C)}{2(1+\exp(C))^2}$. The above inequality can be rewritten as

$$\sum_{t=1}^{T} \|\widehat{f}_t - f_t^*\|_n^2 + \frac{1}{n} \sum_{t=1}^{T} \sum_{i=1}^{n} \frac{1}{C_\alpha} \epsilon_{ti}(\widehat{f}_t(\boldsymbol{x}_{ti}) - f_t^*(\boldsymbol{x}_{ti}))$$

$$+ \frac{\lambda_1}{C_\alpha} \sum_{m=1}^{M} \sqrt{\sum_{t=1}^{T} \|\widehat{f}_{mt}\|_n^2} + \frac{\lambda_2}{C_\alpha} \sum_{m=1}^{M} \sqrt{\sum_{t=1}^{T} \|\widehat{f}_{mt}\|_{\mathcal{H}_m}^2} + \frac{\lambda_3}{C_\alpha} \sum_{m=1}^{M} \sum_{t=1}^{T} \|\widehat{f}_{mt}\|_{\mathcal{H}_m}^2$$

$$\leq \frac{\lambda_1}{C_\alpha} \sum_{m\in I_0} \sqrt{\sum_{t=1}^{T} \|f_{mt}^*\|_n^2} + \frac{\lambda_2}{C_\alpha} \sum_{m\in I_0} \sqrt{\sum_{t=1}^{T} \|f_{mt}^*\|_{\mathcal{H}_m}^2} + \frac{\lambda_3}{C_\alpha} \sum_{m\in I_0} \sum_{t=1}^{T} \|f_{mt}^*\|_{\mathcal{H}_m}^2.$$

Applying the following inequalities

$$\sum_{m\in I_0} \sqrt{\sum_{t=1}^{T} \|f_{mt}^*\|^2} - \sum_{m\in I_0} \sqrt{\sum_{t=1}^{T} \|\widehat{f}_{mt}\|^2} \leq \sum_{m\in I_0} \sqrt{\sum_{t=1}^{T} \|\widehat{f}_{mt} - f_{mt}^*\|^2}$$

and

$$\|f_{mt}^*\|_{\mathcal{H}_m}^2 - \|\widehat{f}_{mt}\|_{\mathcal{H}_m}^2 = 2\langle f_{mt}^*, f_{mt}^* - \widehat{f}_{mt}\rangle_{\mathcal{H}_m} - \|\widehat{f}_{mt} - f_{mt}^*\|_{\mathcal{H}_m}^2,$$

we have

$$\sum_{t=1}^{T} \|\widehat{f}_t - f_t^*\|_n^2 + \frac{1}{n} \sum_{t=1}^{T} \sum_{i=1}^{n} \frac{1}{C_\alpha} \epsilon_{ti}(\widehat{f}_t(\boldsymbol{x}_{ti}) - f_t^*(\boldsymbol{x}_{ti}))$$

$$+ \frac{\lambda_1}{C_\alpha} \sum_{m\in I_0^c} \sqrt{\sum_{t=1}^{T} \|\widehat{f}_{mt}\|_n^2} + \frac{\lambda_2}{C_\alpha} \sum_{m\in I_0^c} \sqrt{\sum_{t=1}^{T} \|\widehat{f}_{mt}\|_{\mathcal{H}_m}^2} + \frac{\lambda_3}{C_\alpha} \sum_{m\in I_0^c} \sum_{t=1}^{T} \|\widehat{f}_{mt}\|_{\mathcal{H}_m}^2$$

$$\leq \frac{\lambda_1}{C_\alpha} \sum_{m\in I_0} \sqrt{\sum_{t=1}^{T} \|\widehat{f}_{mt} - f_{mt}^*\|_n^2} + \frac{\lambda_2}{C_\alpha} \sum_{m\in I_0} \sqrt{\sum_{t=1}^{T} \|\widehat{f}_{mt} - f_{mt}^*\|_{\mathcal{H}_m}^2}$$

$$+ \frac{\lambda_3}{C_\alpha} \sum_{m\in I_0} \sum_{t=1}^{T} (2\langle f_{mt}^*, f_{mt}^* - \widehat{f}_{mt}\rangle_{\mathcal{H}_m} - \|\widehat{f}_{mt} - f_{mt}^*\|_{\mathcal{H}_m}^2).$$

Thus on the event $\mathcal{E}_2(r)$, by Equation (H.1) and Equation (H.2), dropping the term of $\lambda_3$ on the LHS, we have

$$\sum_{t=1}^{T} \|\widehat{f}_t - f_t^*\|_n^2 + \frac{1}{C_\alpha} \frac{1}{2} \sum_{m\in I_0^c} \left( \lambda_1 \sqrt{\sum_{t=1}^{T} \|\widehat{f}_{mt}\|_{L_2(\Pi)}^2} + \lambda_2 \sqrt{\sum_{t=1}^{T} \|\widehat{f}_{mt}\|_{\mathcal{H}_m}^2} \right)$$

$$\leq \left| \frac{1}{n} \sum_{t=1}^{T} \sum_{i=1}^{n} \frac{1}{C_\alpha} \epsilon_{ti}(\widehat{f}_t(\boldsymbol{x}_{ti}) - f_t^*(\boldsymbol{x}_{ti})) \right| + \frac{\lambda_3}{C_\alpha} \sum_{m\in I_0} \sum_{t=1}^{T} (2\langle f_{mt}^*, f_{mt}^* - \widehat{f}_{mt}\rangle_{\mathcal{H}_m} - \|\widehat{f}_{mt} - f_{mt}^*\|_{\mathcal{H}_m}^2)$$

$$+ \frac{1}{C_\alpha} \frac{3}{2} \sum_{m\in I_0} \left( \lambda_1 \sqrt{\sum_{t=1}^{T} \|\widehat{f}_{mt} - f_{mt}^*\|_{L_2(\Pi)}^2} + \lambda_2 \sqrt{\sum_{t=1}^{T} \|\widehat{f}_{mt} - f_{mt}^*\|_{\mathcal{H}_m}^2} \right).$$

Moreover, on the event $\mathcal{E}_1(t)$, we obtain

$$
\begin{aligned}
&\sum_{t=1}^{T} \|\widehat{f}_t - f_t^*\|_n^2 + \frac{1}{C_\alpha} \frac{1}{2} \sum_{m \in I_0^c} \left( \lambda_1 \sqrt{\sum_{t=1}^{T} \|\widehat{f}_{mt}\|_{L_2(\Pi)}^2} + \lambda_2 \sqrt{\sum_{t=1}^{T} \|\widehat{f}_{mt}\|_{\mathcal{H}_m}^2} \right) \\
&\leq \sum_{m=1}^{M} \frac{1}{C_\alpha} \sqrt{c_\nu \omega_n (T + \sqrt{T} \log M) u} \left( \sqrt{\sum_{t=1}^{T} \|\widehat{f}_{mt} - f_{mt}^*\|_{L_2(\Pi)}^2} + \lambda^{\frac{1}{2}} \sqrt{\sum_{t=1}^{T} \|\widehat{f}_{mt} - f_{mt}^*\|_{\mathcal{H}_m}^2} \right) \\
&\quad + \frac{1}{C_\alpha} \frac{3}{2} \sum_{m \in I_0} \left( \lambda_1 \sqrt{\sum_{t=1}^{T} \|\widehat{f}_{mt} - f_{mt}^*\|_{L_2(\Pi)}^2} + \lambda_2 \sqrt{\sum_{t=1}^{T} \|\widehat{f}_{mt} - f_{mt}^*\|_{\mathcal{H}_m}^2} \right) \\
&\quad + \frac{\lambda_3}{C_\alpha} \sum_{m \in I_0} \sum_{t=1}^{T} (2\langle f_{mt}^*, f_{mt}^* - \widehat{f}_{mt} \rangle_{\mathcal{H}_m} - \|\widehat{f}_{mt} - f_{mt}^*\|_{\mathcal{H}_m}^2).
\end{aligned} \tag{H.3}
$$

Using the relation $4\sqrt{c_\nu \omega_n (T + \sqrt{T} \log M) u} = \lambda_1$ and $\lambda_2 = \lambda_1 \lambda^{\frac{1}{2}}$ and splitting the first term on the RHS, the above inequality yields

$$
\begin{aligned}
&\frac{1}{4} \sum_{m \in I_0^c} \left( \lambda_1 \sqrt{\sum_{t=1}^{T} \|\widehat{f}_{mt}\|_{L_2(\Pi)}^2} + \lambda_2 \sqrt{\sum_{t=1}^{T} \|\widehat{f}_{mt}\|_{\mathcal{H}_m}^2} \right) \\
&\leq \sum_{m \in I_0} \frac{7}{4} \left( \lambda_1 \sqrt{\sum_{t=1}^{T} \|\widehat{f}_{mt} - f_{mt}^*\|_{L_2(\Pi)}^2} + \lambda_2 \sqrt{\sum_{t=1}^{T} \|\widehat{f}_{mt} - f_{mt}^*\|_{\mathcal{H}_m}^2} \right) + \sum_{m \in I_0} 2\lambda_3 \sum_{t=1}^{T} \langle f_{mt}^*, f_{mt}^* - \widehat{f}_{mt} \rangle_{\mathcal{H}_m},
\end{aligned}
$$

where we remove the first term on the LHS, the last term ($\mathcal{H}_m$-norm) on the RHS, and $C_\alpha$ on both sides. By the Cauchy-Schwarz inequality, we have

$$
\begin{aligned}
\lambda_3 \sum_{t=1}^{T} \langle f_{mt}^*, f_{mt}^* - \widehat{f}_{mt} \rangle_{\mathcal{H}_m} &\leq \lambda_3^{\frac{1}{2}} \sum_{t=1}^{T} \|f_{mt}^*\|_{\mathcal{H}_m} \sqrt{\|\widehat{f}_{mt} - f_{mt}^*\|_{L_2(\Pi)}^2 + \lambda_3 \|\widehat{f}_{mt} - f_{mt}^*\|_{\mathcal{H}_m}^2} \\
&\leq \lambda_3^{\frac{1}{2}} \sqrt{\sum_{t=1}^{T} \|f_{mt}^*\|_{\mathcal{H}_m}^2} \sqrt{\sum_{t=1}^{T} \|\widehat{f}_{mt} - f_{mt}^*\|_{L_2(\Pi)}^2 + \sum_{t=1}^{T} \lambda_3 \|\widehat{f}_{mt} - f_{mt}^*\|_{\mathcal{H}_m}^2} \\
&\leq \lambda_3^{\frac{1}{2}} \sqrt{\sum_{t=1}^{T} \|f_{mt}^*\|_{\mathcal{H}_m}^2} \left( \sqrt{\sum_{t=1}^{T} \|\widehat{f}_{mt} - f_{mt}^*\|_{L_2(\Pi)}^2} + \lambda_3^{\frac{1}{2}} \sqrt{\sum_{t=1}^{T} \|\widehat{f}_{mt} - f_{mt}^*\|_{\mathcal{H}_m}^2} \right),
\end{aligned} \tag{H.4}
$$

where in the second inequality we use the Cauchy-Schwarz inequality. Therefore we obtain

$$
\begin{aligned}
&\frac{1}{4} \sum_{m \in I_0^c} \left( \lambda_1 \sqrt{\sum_{t=1}^{T} \|\widehat{f}_{mt}\|_{L_2(\Pi)}^2} + \lambda_2 \sqrt{\sum_{t=1}^{T} \|\widehat{f}_{mt}\|_{\mathcal{H}_m}^2} \right) \\
&\leq \sum_{m \in I_0} \frac{7}{4} \left( \lambda_1 \sqrt{\sum_{t=1}^{T} \|\widehat{f}_{mt} - f_{mt}^*\|_{L_2(\Pi)}^2} + \lambda_2 \sqrt{\sum_{t=1}^{T} \|\widehat{f}_{mt} - f_{mt}^*\|_{\mathcal{H}_m}^2} \right) \\
&\quad + \sum_{m \in I_0} 2\lambda_3^{\frac{1}{2}} \sqrt{\sum_{t=1}^{T} \|f_{mt}^*\|_{\mathcal{H}_m}^2} \left( \sqrt{\sum_{t=1}^{T} \|\widehat{f}_{mt} - f_{mt}^*\|_{L_2(\Pi)}^2} + \lambda_3^{\frac{1}{2}} \sqrt{\sum_{t=1}^{T} \|\widehat{f}_{mt} - f_{mt}^*\|_{\mathcal{H}_m}^2} \right),
\end{aligned}
$$

on the events $\mathcal{E}_1(u)$ and $\mathcal{E}_2(r)$. Since $\|f_{mt}^*\|_n^2 \geq \|f_{mt}^* - \widehat{f}_{mt}\|_n^2$ and $\|\widehat{f}_{mt}\|_{L_2(\Pi)}^2 \geq \|\widehat{f}_{mt} - f_{mt}^*\|_{L_2(\Pi)}^2$, by adding $\frac{1}{4} \sum_{m \in I_0} \left( \lambda_1 \sqrt{\sum_{t=1}^{T} \|\widehat{f}_{mt}\|_{L_2(\Pi)}^2} + \lambda_2 \sqrt{\sum_{t=1}^{T} \|\widehat{f}_{mt}\|_{\mathcal{H}_m}^2} \right)$ to both sides, we get the assertion (E.2). $\qquad \square$