# OpenReview forum: "CLIP as Multi-Task Multi-Kernel Learning"
_ICLR.cc/2024/Conference — Submitted to ICLR 2024_

### Official Review · Reviewer_ry4r · 2023-10-27

**Soundness:** 2 fair
**Presentation:** 3 good
**Contribution:** 3 good
**Rating:** 6
**Confidence:** 3

**Summary:**

This paper casts the problem of estimating high dimensional neural networks mappings as selecting an unknown Reproducible Kernel Hilbert Space (RKHS) using the optimal solution of a multi-task multiple kernel learning (MTMKL) optimization problem. Under the setting where both number of covariates and the number of candidate kernels increase with the sample size, the authors show an optimal statistical rate of the MTMKL classifier. The proposed method is successfully applied to embeddings of medical imaging data.

**Strengths:**

- Casting the high dimensional neural networks optimization problem as multi-task multiple learning optimization problem
- Thorough theoretical analysis of the optimal estimator

**Weaknesses:**

- CLIP is not considered in the experiments! This paper should not be sold as CLIP modelling but as a method that use inner-product based objective.
- Claim that there were no prior theoretical work on multi-task multiple kernel learning. There are multiple work on the subject (this is an old topic) including theoretical analysis. See (Micchelli, C., & Pontil, M. (2004). Kernels for Multi--task Learning. Advances in neural information processing systems, 17.)

**Questions:**

- I have a big concern about the title of the paper. I do not see why the results in the paper are only specialized to CLIP models instead of general estimating functionals of high dimensional (features embeddings) inner products. In the experimental section, CLIP is barely used! Some embeddings are just extracted. I think if the paper is around CLIP as Multi-task multiple kernel, there should be a whole study on the architecture of CLIP, which to me seems to be a big task. Could the authors please elaborate more on this as this is really confusing to me about the real contribution of the paper?
-It is mentioned in the Related literature that no theoretical analysis of MKL has been conductied in the multi-task setting. Could the authors justify that as there is a big literature on the subject. For instance: See (Micchelli, C., & Pontil, M. (2004). Kernels for Multi--task Learning. Advances in neural information processing systems, 17.) which has not been cited/analysed. There are other related works. Could the authors elaborate on this?
- In the experiments section (section 7.2), the authors do not specify which embeddings are used (are they CLIP embeddings?), justifying again my concerns of why CLIP as a motivation of the paper? If so, more detailed experiments should be performed using CLIP embeddings for large-scale image recognition problem (this is a big literature to be considered).


------- After rebuttal ---------------------
The work conducted by the authors during the rebuttal phase convinced me to raise my score. My original concern was that the paper was focused on proposing a new way of training CLIP-like models but the experimental sections did not compare with respect to the traditional way of training CLIP. The authors have done this comparison in the rebuttal phase and have shown improvements of their MTMK approach over CLIP, and promised to revise the paper to include that as one of the message of the paper by including it as an algorithm for training CLIP.

---

> ### Author Response · Authors · 2023-11-13
>
> > Q1: I do not see why the results in the paper are only specialized to CLIP models instead of general estimating functionals of high dimensional (features embeddings) inner products, ..., Could the authors please elaborate more on this as this is really confusing to me about the real contribution of the paper?
>
> Thank you for the question. We appreciate the comment about the interpretation of our method beyond CLIP as general estimating functionals of high-dimensional inner products. Nonetheless, the main motivation for the proposed method is to provide an alternative method to solve the CLIP loss and some theoretical understanding of CLIP from the RKHS perspective instead of the neural network. We acknowledge the complexity of thoroughly analyzing the intricate neural architecture of CLIP, which is why we have focused on solving the loss function of CLIP and attempted to give an alternative interpretation of the essence of CLIP via the kernel method. Therefore, we believe our proposed MTMK method should be considered a novel methodological contribution to the CLIP community.
>
> > Q2: It is mentioned in the Related literature that no theoretical analysis of MKL has been conductied in the multi-task setting. Could the authors justify that as there is a big literature on the subject?
>
> We appreciate your concern and the reference provided. We apologize for the imprecise wording and the lack of citations in our initial submission. While there is indeed a substantial body of literature discussing kernel constructions, optimization, and computation aspects of MKL in the multi-task setting, we intended to highlight the scarcity of works that specifically analyze the optimal learning rate in multi-task multi-kernel learning. The statement "no theoretical analysis" refers to the limited research on the analysis of the statistical rate. In the revised version of our paper, we will include a discussion on the general theoretical research line in multi-task multi-kernel learning, along with relevant citations such as [1-6]. We will also ensure that the wording is accurate and precise.
>
> [1] Micchelli, C., & Pontil, M. (2004). Kernels for Multi--task Learning. *Advances in neural information processing systems*, *17*.
>
> [2] Dinuzzo, F., Ong, C. S., Pillonetto, G., & Gehler, P. V. (2011). Learning output kernels with block coordinate descent. In *Proceedings of the 28th International Conference on Machine Learning (ICML-11)* (pp. 49-56).
>
> [3] Ciliberto, C., Mroueh, Y., Poggio, T., & Rosasco, L. (2015, June). Convex learning of multiple tasks and their structure. In *International Conference on Machine Learning* (pp. 1548-1557). PMLR.
>
> [4] Ciliberto, C., Rosasco, L., & Villa, S. (2015). Learning multiple visual tasks while discovering their structure. In *Proceedings of the IEEE Conference on Computer Vision and Pattern Recognition* (pp. 131-139).
>
> [5] Jawanpuria, P. K., Lapin, M., Hein, M., & Schiele, B. (2015). Efficient output kernel learning for multiple tasks. *Advances in neural information processing systems*, *28*.
>
> [6] Pentina, A., & Ben-David, S. (2015). Multi-task and lifelong learning of kernels. In *Algorithmic Learning Theory: 26th International Conference, ALT 2015, Banff, AB, Canada, October 4-6, 2015, Proceedings 26* (pp. 194-208). Springer International Publishing.
>
> > Q3: In the experiments section (section 7.2), the authors do not specify which embeddings are used (are they CLIP embeddings?), justifying again my concerns of why CLIP as a motivation of the paper?
>
> Thank you for the question. We perceive $\phi(x)$ as the embedding of $x$ and derive the corresponding inner products between samples via the method described in Section 4.  Therefore, the embedding derived for images and phecodes in Section 7.2 corresponds to the estimated mapping of our MTMK method. As for the initial word embeddings for phecodes, we use the ones provided in the MIKGI dataset described in the third paragraph.
>
> > Q4: More CLIP related experiments?
>
> Thanks for the question. To further illustrate the performance of the neural network-based CLIP, we will conduct some new experiments to further showcase the empirical performance. We hope the additional results (to be uploaded) resolve your concerns.

---

> > ### Comment · Reviewer_ry4r · 2023-11-20
> >
> > Thanks to the authors for giving clarification on the questions and concerns. I still believe the scope/message of the paper needs to be changed as the story/theory and practical experiments are not really aligned.

---

### Official Review · Reviewer_F8DR · 2023-11-01

**Soundness:** 3 good
**Presentation:** 3 good
**Contribution:** 2 fair
**Rating:** 6
**Confidence:** 3

**Summary:**

The paper tries to achieve better understanding and analysis of CLIP using reproducing kernel Hilbert spaces (RKHS). CLIP uses a contrastive loss on mapping of input text and image datasets. The paper argues that the objective function of CLIP can be expressed by using kernels in an RKHS. Then the paper proposes to find the best RKHS that maximizes the contrastive loss of CLIP.

**Strengths:**

1. Interpreting the training CLIP as a kernel learning problem is interesting. Due to characteristics of kernel functions analyzing kernel-based models is tractable. This perspective offers a promising avenue for enhancing the understanding of CLIP, a fundamental and widely adopted model.
2. The paper conducted experiments on real datasets to support the theoretical analysis.

**Weaknesses:**

1. It seems that the goal of the paper is to give readers of better understanding on how to train a better CLIP model. It is more common that neural network models are employed. However, I did not find any discussion in the paper that if their analysis provides some intuition on how  train a better CLIP. Therefore, the contribution of this paper is not super clear to me.
2. The methods used in this paper is too heuristics. In case of using neural networks, deep neural network models can be interpreted as NTK using some heuristics while in order to train the contrastive loss using kernel learning, the paper performs some relaxations in the objective function.
3. Experimental section can be improved by adding more datasets and baselines.

**Questions:**

Please see weaknesses above.

---

> ### Author Response · Authors · 2023-11-13
>
> Thanks for the constructive comments, we address the concerns below.
>
> > Q1: How do we interpret the contribution of this work?
>
> Thank you for the question. We expect that our work offers some theoretical understanding of CLIP from the RKHS perspective rather than the standard neural network perspective. In essence, we do not aim at proposing a new method to train a better CLIP model. Instead, we attempt to provide an alternative method to solve the CLIP loss, which is parallel to the standard backpropagation training scheme from a higher level. Therefore, we believe our proposed MTMK method should be considered a novel methodological contribution to the CLIP community instead of a remedy to the existing limitations of CLIP.
>
> > Q2: The methods used in this paper is too heuristics.
>
> We appreciate the reviewer for bringing up this concern. We acknowledge that there is a certain degree of relaxation between the original loss and the objective we have derived. However, we argue that such relaxation is necessary to statistically characterize the complexity and learning rate of the model, as presented in Section 6.
>
> > Q3: Experimental section can be improved by adding more datasets and baselines.
>
> We appreciate your suggestion on improving the numerical evaluation. However, we would like to emphasize that this work mainly focuses on a theoretical interpretation of CLIP rather than proposing a new method. As CLIP has demonstrated excellent generalizability in broader domains, we believe our work can be extended to various tasks as CLIP. To further illustrate the performance of the neural network-based CLIP, we will conduct some new experiments to further showcase the empirical performance. We hope the additional results (to be uploaded) resolve your concerns.

---

### Official Review · Reviewer_pvdu · 2023-11-28

**Soundness:** 2 fair
**Presentation:** 2 fair
**Contribution:** 2 fair
**Rating:** 3
**Confidence:** 3

**Summary:**

In this paper, the authors first derive relations between CLIP and multi-task multi kernel (MTMK) learning. Then, using the established relations, they discuss MTMK Logistic regression, where they also discuss on the various regularization choices such as L1, L2, Group Lasso and combination of them. They derive the conditions for the consistency of the estimator.
The numerical experiments show the proposed schemes (involving various combinations of the regularizers) achieve better performance in synthetic and real datasets in contrast to baselines such as SVM and LR.

**Strengths:**

- In Section 6, The theoretical study of the convergence of the estimators under multi task multi kernel setup is interesting.
- It is an interesting attempt to study the relations between the kernel methods to the pretraining steps in CLIP

**Weaknesses:**

- It is not clear about the focus of the paper. While the primary motivation has been understanding CLIP, the latter part of the paper doesn't offer any discussions, neither there are any experimental comparisons to that of CLIP.

- It was difficult (for the reviewer) to understand and comprehend the main equivalances. It would be great to have clarifications or if the writing is improved to clear the confusions. Examples below:
1. The notation itself is a bit confusing to understand. While CLIP uses pseudolabels, which have n different values for n samples, this paper has a notation T for the classes and n for the samples. It would be easier to have a clear writeup on the dataset construction before proceeding to the next equivalances.

2. It is not very clear how equation 2.1 represents CLIP. Since in CLIP, the inner product is computed between a text representation of sample i and image representation of sample j. Here, it is assumed that \phi captures both. It would be nice to provide a concrete reference to show that CLIP objective is max_H C^D_H, as mentioned in the paper.


3. It is difficult to understand the correlate the final reduction in page 5 (the equation on the top) to that of CLIP. Because CLIP labels are defined for pairs, and here the label is defined for a sample.

- The experimental section is not in line with the initial claims of the paper.

**Questions:**

- What happens if we assume that the CLIP objective (2.1) has dot(\phi_I(x_i), \phi_t(x_j)) corresponding to the image and text embeddings. Are the results discussed in this paper hold without any loss of generality ?

- In 2.7, Is y_ti one hot encoding of the ith sample ?

- It might be better to read if better rigour had been followed, for instance in defining what category the loss function f belongs to. The equations 2.2-2.7 mention maximization with a loss function, while it is changed to min at the end. Probably this is a typo.

- In the synthetic experiments, more explanation might be needed on the ground truth function. And some explanation would be needed why the MTMK-L1 worked best in relation to the true model.

- Since the model is motivated from CLIP, how do we compare against tasks which are used for evaluating CLIP

---

> ### Author Response · Authors · 2023-11-29
>
> Thanks for the constructive comments. As an overview, we expect that our work provides an alternative tool to standard training schemes of neural networks with some novel insight from the RKHS perspective. Now we address the concerns below.
>
> > Q1: It is not clear about the focus of the paper. While the primary motivation has been understanding CLIP, the latter part of the paper doesn't offer any discussions, ...
>
> Thank you for the question. The main focus of the paper is to provide an alternative method to solve the CLIP loss and thereby some theoretical understanding of CLIP from the RKHS perspective instead of the neural network. We acknowledge that we focus more on the theoretical analysis of MTMK in the latter parts while the discussions are mainly offered in Section 2. The reason why we organized in this fashion is that we believe once the equivalence between standard CLIP and our proposed method is built, the theoretical analysis (Section 6) and experiments (Section 7) can be applied to CLIP from a theoretical viewpoint. In the revised version, we will include a more comprehensive discussion in later sections.
>
> > Q2: The notation itself is a bit confusing to understand. While CLIP uses pseudolabels, which have n different values for n samples, this paper has a notation T for the classes and n for the samples. It would be easier to have a clear writeup on the dataset construction before proceeding to the next equivalances.
>
> Thank you for the comment. The $T$ used in this paper refers to the classes that appeared in the whole dataset for ease of analysis of the learning rate. In the revised version, we will include further clarification with regard to the dataset construction.
>
> > Q3: It is not very clear how equation 2.1 represents CLIP. Since in CLIP, the inner product is computed between a text representation of sample i and image representation of sample j. Here, it is assumed that \phi captures both. It would be nice to provide a concrete reference to show that CLIP objective is max_H C^D_H, as mentioned in the paper.
>
> Thank you for the comment. We use the same $\phi$  for notation simplicity in the following analysis, and extending to different $\phi$  for texts and images should be straightforward (for example, using an indicator function to indicate whether it is a text or image sample). Thus, it should be clear now that texts and images are pushed through $\phi$  separately. We will include the above clarification in the revised caption. In addition, equation (2.1) is just the mathematical abstraction of the original CLIP's idea, which is "To do this, CLIP learns a multi-modal embedding space by jointly training an image encoder and text encoder to maximize the cosine similarity of the image and text embeddings of the $N$ real pairs in the batch while minimizing the cosine similarity of the embeddings of the $N^2 − N$ incorrect pairings." [1]
>
> [1] Radford, A., Kim, J. W., Hallacy, C., Ramesh, A., Goh, G., Agarwal, S., ... & Sutskever, I. (2021, July). Learning transferable visual models from natural language supervision. In *International conference on machine learning* (pp. 8748-8763). PMLR.
>
> > Q4: It is difficult to understand the correlate the final reduction in page 5 (the equation on the top) to that of CLIP. Because CLIP labels are defined for pairs, and here the label is defined for a sample.
>
> Thanks for raising this question. Given the complexity of the theoretical analysis of learning rate, we assume the pseudo labels of sample pairs used by CLIP can be treated as labels in our analysis (indicated on page 3 below equation (2.1)). Therefore, in our notation, for a sample pair $\boldsymbol{x}_i, \boldsymbol{x}_j, \forall i,j \in [N]$, $y_i = y_j$ denote a correct pairing while $y_i \ne y_j$ denote an incorrect pairing.
>
> > Q5: ..., neither there are any experimental comparisons to that of CLIP. The experimental section is not in line with the initial claims of the paper.
>
> Thanks for the question. To further illustrate the performance of the neural network-based CLIP, we conducted some new experiments to showcase the empirical performance. We hope the additional results in the official comments resolve your concerns.
>
> > Q6: What happens if we assume that the CLIP objective (2.1) has dot(\phi_I(x_i), \phi_t(x_j)) corresponding to the image and text embeddings. Are the results discussed in this paper hold without any loss of generality ?
>
> The results hold without any loss of generality as we can directly write $\phi(x) = \mathbb{I}\\{x ~ \text{is an image}\\}\phi_I(x) + \mathbb{I}\\{x ~ \text{is a text}\\}\phi_t(x)$.
>
> > Q7: In 2.7, Is y_ti one hot encoding of the ith sample ?
>
> We apologize for the confusion. In line with $y_i$, $y_{ti} \in \\{+1, -1\\}.$

---

> > ### Author Response · Authors · 2023-11-29
> >
> > > Q8: It might be better to read if better rigour had been followed, for instance in defining what category the loss function f belongs to. The equations 2.2-2.7 mention maximization with a loss function, while it is changed to min at the end. Probably this is a typo.
> >
> > Thank you for pointing out. This is a typo, we will fix this issue in the revised version.
> >
> > > Q9: In the synthetic experiments, more explanation might be needed on the ground truth function. And some explanation would be needed why the MTMK-L1 worked best in relation to the true model.
> >
> > Thank you the question. In the synthetic experiments, the detailed task configuration is provided in Appendix C, where each ground truth function can be represented via a sparse combination of the candidate kernels (e.g., in the task set 1, the truth is a combination of polynomials of order 1, 3 and 5). Thus, the candidate kernels consist of ones with a positive effect and ones with a negative effect and we need to correctly select the useful kernels. Our algorithm is specifically designed for this scenario and is capable of selecting the best kernels with the L1 penalty. Furthermore, we provide a theoretical guarantee to depict how our proposed method can correctly select the kernels.
> >
> > > Q10: Since the model is motivated from CLIP, how do we compare against tasks which are used for evaluating CLIP.
> >
> > Thank you for the question. Our method can mainly be used on classification tasks aligned with CLIP. However, we would like to emphasize that this work mainly focuses on a theoretical interpretation of CLIP rather than proposing a new method.

---

> ### Author Response · Authors · 2023-12-01
> **Follow Up Reminder to Reviewer pvdu**
>
> Dear Reviewer pvdu,
>
> As the author-reviewer discussion period ends soon, we would appreciate it if you could check our response to your review comments. This way, if you have further questions and comments, we can still reply before the author-reviewer discussion period ends. If our response resolves your concerns, we kindly ask you to consider raising the rating of our work. Thank you very much for your time and efforts!

---

> > ### Comment · Reviewer_pvdu · 2023-12-01
> > **Thanks for the clarifications**
> >
> > Dear authors,
> > Thank you very much for the clarifications.
> > - On the explanations on equivalance, while the intuitive explanations help, since one of the main focus of the papers is the equivalance between CLIP and MTML, it would help with more rigourus derivations.
> > For instance, when we assume that the \phi(x) is defined as summation expression (answered for Q6), it does not directly imply how we encode negative samples. an image+text pair which is correct has a positive label, and image+text pair which has wrong pairing has a negative label. A rigorous derivation would help clear the confusions.
> >
> > - On the experiments front, since the focus is towards the equivalance, it would be helpful for verifying the equivalances through direct comparisons on the same tasks.
> > Looking forward for a revised version incorporating the clarifications. Best wishes!

---

> > > ### Author Response · Authors · 2023-12-01
> > >
> > > > Q1: On the explanations on equivalance, while the intuitive explanations help, since one of the main focus of the papers is the equivalance between CLIP and MTML, it would help with more rigourus derivations, ..., On the experiments front, since the focus is towards the equivalance, it would be helpful for verifying the equivalances through direct comparisons on the same tasks.
> > >
> > > Thank you for the question. While we acknowledge that on of the main focus of this paper is the connection between standard CLIP and MTMK, we emphasize that we essentially propose an **alternative** method to solve the CLIP loss (which shares the same optimal solution), instead of analyzing the intricate neural architecture of the original CLIP. Notice that neither the standard CLIP nor our method can get the absolute optimal solution, as each method focuses on a specific hypothesis class (standard CLIP is the function space of neural networks, and our MTMK is (2.3)). Therefore, we suppose the notion of complete equivalence is a misunderstanding of our contribution. Nonetheless, we appreciate your valuable comments and would love to make further clarification in the revised version. (However, we are not able to upload a revised version now due to the passed deadline.)
> > >
> > > > Q2: For instance, when we assume that the \phi(x) is defined as summation expression (answered for Q6), it does not directly imply how we encode negative samples. an image+text pair which is correct has a positive label, and image+text pair which has wrong pairing has a negative label. A rigorous derivation would help clear the confusions.
> > >
> > > Thank you for raising this question. In our notation, $\phi(x)$ is only the embedding of $x$ which is exactly the same as the original CLIP. As for the label issue, we use $(\mathbb I\\{y_i = y_j\\} − \mathbb I \\{y_i \ne y_j\\})$ (as in Equation (2.1)) to denote the positive/negative pairing, which should be rigorous in describing the pairing relationship.
> > >
> > > We hope the answers above resolve your concerns and thanks for your time and commitment.

---

### Author Response · Authors · 2023-11-21
**Supplementary experiments 1**

# Rebuttal Experiment

## 1. Fine-tuned CLIP Model

In our study, we utilize image data from the MedMNIST dataset and word embeddings from the MIKGI dataset to assess the efficacy of our proposed
model.

To further demonstrate the efficiency of our approach, we integrate a pretrained CLIP model as an additional step. Specifically, we select [openai/clip-vit-base-patch16](https://huggingface.co/openai/clip-vit-base-patch16) as our backbone model for the image encoder. We input our originalimages into this image encoder to generate image embeddings. Furthermore, we incorporate a learned linear matrix to map our predefined text embeddings to a space with the same dimension as the image embeddings. The process of obtaining our final text embeddings and image embeddings is illustrated in this [diagram](https://ibb.co/sHJ5dpr).

To fine-tune the image encoder and train our linear matrix, we create 1200 matched and mismatched text-image pairs with data from MedMNIST and
MIKGI, employing cross entropy as the criterion.

After completing these steps, we employ a logistic regression model to fit the multi-task dataset, with the only variation being the embeddings compared to our previous experiments. Presented below are the results, along with our previous findings:

| Model    | Average             | Image               | Text                |
| -------- | ------------------- | ------------------- | ------------------- |
| MTMK mix | **0.9074 ± 0.0156** | 0.9310 ± 0.0175     | 0.8973 ± 0.0310     |
| MTMK L1  | *0.9102 ± 0.0129*   | **0.9357 ± 0.0131** | 0.9003 ± 0.0278     |
| MTMK L2  | 0.9050 ± 0.0172     | 0.9278 ± 0.0202     | 0.8973 ± 0.0306     |
| MTSK     | 0.8870 ± 0.0121     | 0.9104 ± 0.0135     | 0.8630 ± 0.0261     |
| STMK     | 0.9053 ± 0.0142     | *0.9366 ± 0.0128*   | 0.8920 ± 0.0339     |
| SVM      | 0.8267 ± 0.0179     | 0.9188 ± 0.0235     | 0.8380 ± 0.0379     |
| LR       | 0.8879 ± 0.0110     | 0.8962 ± 0.0112     | 0.8970 ± 0.0307     |
| CLIP     | 0.7986 ± 0.0264     | 0.7608 ± 0.0133     | **0.9417 ± 0.0356** |

---

> ### Author Response · Authors · 2023-11-21
> **Supplementary experiments 2**
>
> ## 2. Choose Different Kernels
>
> Expanding from the foundational task set, we select three task groups, each encompassing six distinct tasks. Within these task sets, each individual task is associated with 1000 samples. Four sets of experiments were conducted to evaluate performance, utilizing the average mean squared error and standard deviation across five different random seeds for validation.
>
> 1. **Experiment 1: Employing the original experiment's kernel list as the control group**
>
>    | Model    | Task Set 1          | Task Set 2          | Task Set 3          |
>    | -------- | ------------------- | ------------------- | ------------------- |
>    | ORACLE   | 0.0107 ± 0.0030     | 0.0159 ± 0.0035     | 0.0257 ± 0.0148     |
>    | MTMK mix | *0.0201 ± 0.0135*   | *0.0310 ± 0.0135*   | *0.0422 ± 0.0150*   |
>    | MTMK L1  | **0.0193 ± 0.0128** | **0.0259 ± 0.0116** | **0.0382 ± 0.0136** |
>    | MTMK L2  | 0.0251 ± 0.0141     | 0.0267 ± 0.0108     | 0.0504 ± 0.0155     |
>    | MTSK     | 0.1355 ± 0.0110     | 0.1295 ± 0.0115     | 0.3064 ± 0.0260     |
>    | STMK     | 0.0245 ± 0.0120     | 0.0296 ± 0.0117     | 0.0554 ± 0.0112     |
>    | STSK     | 0.1381 ± 0.0116     | 0.1309 ± 0.0081     | 0.3055 ± 0.0239     |
>
> 2. **Experiment 2: Introducing three additional Neural Tangent Kernels**
>
> | Model    | Task Set 1          | Task Set 2          | Task Set 3          |
> | -------- | ------------------- | ------------------- | ------------------- |
> | ORACLE   | 0.0107 ± 0.0030     | 0.0159 ± 0.0035     | 0.0257 ± 0.0148     |
> | MTMK mix | *0.0222 ± 0.0143*   | *0.0291 ± 0.0129*   | *0.0428 ± 0.0148*   |
> | MTMK L1  | **0.0222 ± 0.0133** | **0.0254 ± 0.0106** | **0.0384 ± 0.0140** |
> | MTMK L2  | 0.0249 ± 0.0141     | 0.0290 ± 0.0129     | 0.0510 ± 0.0143     |
> | MTSK     | 0.1355 ± 0.0110     | 0.1295 ± 0.0115     | 0.3064 ± 0.0260     |
> | STMK     | 0.0289 ± 0.0145     | 0.0345 ± 0.0137     | 0.0441 ± 0.0187     |
> | STSK     | 0.1381 ± 0.0116     | 0.1309 ± 0.0081     | 0.3055 ± 0.0239     |
>
> 3. **Experiment 3: Misspecified Models excluding the oracle kernels, in contrast to Experiment 1**
>
>    | Model    | Task Set 1          | Task Set 2          | Task Set 3          |
>    | -------- | ------------------- | ------------------- | ------------------- |
>    | ORACLE   | 0.0107 ± 0.0030     | 0.0159 ± 0.0035     | 0.0257 ± 0.0148     |
>    | MTMK mix | *0.0245 ± 0.0145*   | *0.0323 ± 0.0134*   | *0.0410 ± 0.0159*   |
>    | MTMK L1  | **0.0213 ± 0.0122** | 0.0308 ± 0.0140     | **0.0390 ± 0.0164** |
>    | MTMK L2  | 0.0277 ± 0.0143     | **0.0291 ± 0.0108** | 0.0516 ± 0.0182     |
>    | MTSK     | 0.1355 ± 0.0110     | 0.1295 ± 0.0115     | 0.3064 ± 0.0260     |
>    | STMK     | 0.0314 ± 0.0157     | 0.0387 ± 0.0143     | 0.0405 ± 0.0212     |
>    | STSK     | 0.1381 ± 0.0116     | 0.1309 ± 0.0081     | 0.3055 ± 0.0239     |
>
> 4. **Experiment 4: Misspecified Models removing oracle kernels and introducing new NTKs**
>
>    | Model    | Task Set 1          | Task Set 2          | Task Set 3          |
>    | -------- | ------------------- | ------------------- | ------------------- |
>    | ORACLE   | 0.0107 ± 0.0030     | 0.0159 ± 0.0035     | 0.0257 ± 0.0148     |
>    | MTMK mix | *0.0257 ± 0.0153*   | *0.0306 ± 0.0128*   | *0.0422 ± 0.0163*   |
>    | MTMK L1  | **0.0230 ± 0.0137** | **0.0265 ± 0.0108** | **0.0357 ± 0.0147** |
>    | MTMK L2  | 0.0306 ± 0.0145     | 0.0298 ± 0.0132     | 0.0561 ± 0.0172     |
>    | MTSK     | 0.1355 ± 0.0110     | 0.1295 ± 0.0115     | 0.3064 ± 0.0260     |
>    | STMK     | 0.0341 ± 0.0150     | 0.0366 ± 0.0154     | 0.0439 ± 0.0230     |
>    | STSK     | 0.1381 ± 0.0116     | 0.1309 ± 0.0081     | 0.3055 ± 0.0239     |

---

> ### Comment · Reviewer_ry4r · 2023-11-21
>
> Thanks to the authors for performing these additional experiments to show that the proposed method perform better overall compared to the traditional way of training CLIP.
> Questions:
> 1- What is the computational complexity of the MTMK method compared to the training of CLIP you have performed?
> 2- I would stress more as message in the paper that you are proposing a new way of training CLIP and a section on how to do that exactly in the form of an algorithm

---

> > ### Author Response · Authors · 2023-11-21
> >
> > > Q1: What is the computational complexity of the MTMK method compared to the training of CLIP you have performed?
> >
> > Thanks for the question. Our MTMK method is solved via Block Coordinate Gradient Descent with analytical updates, and the time complexity for a single step is $O(n^3 T)$ as determined by the inverse of a diagonal block matrix $\boldsymbol{W}_m$ (This can be further lowered as we do not need to compute the inverse in every iteration, i.e., the complexity is roughly quadratic w.r.t. $n$). Notice that the computational complexity of the kernel method is independent of the embedding dimension and only relies on sample size $n$ and task size $T$. As for the traditional CLIP, we use a pretrained model with a ViT backbone whose parameters are about 150M. In comparison, the computational complexity of forward propagation and backpropagation relies on the embedding dimension, sample size, and task size, which is much greater than our method due to the neural network architecture and the standard training schemes.
> >
> > > Q2: I would stress more as message in the paper that you are proposing a new way of training CLIP and a section on how to do that exactly in the form of an algorithm
> >
> > Thank you for the suggestion. In the revised version of the paper, we will emphasize that our method presents a novel approach to training CLIP. We will highlight this aspect as a key contribution of our research.  Furthermore, we will elaborate on the method section (Section 4) and include a detailed algorithmic description illustrating the step-by-step process of obtaining the desired inner product of embeddings. This will ensure that readers have a clear understanding of the exact procedure involved in training CLIP with our approach.

---

> > > ### Comment · Reviewer_ry4r · 2023-11-21
> > >
> > > Thank you for the clarifications.

---

### Meta-Review · Area_Chair_8sDu · 2023-12-12

**Metareview:**

The paper proposes to use RKHS to previously proposed CLIP, a foundation  model involving both images and texts. It proposes a contrastive learning based loss function involving Kernel functions thus establishing the connection to MultiTask Multiple Kernel (MTMK) learning. This allows the paper to adapt the results in MTMK to CLIP which leads to  an alternative for learning CLIP.  There was consensus that the paper requires more focussed presentation, including a more detailed experimentation, to bring out the advantages of the proposed method.

**Justification For Why Not Higher Score:**

The reviewers have substantial concerns that the paper is under-developed, both in the formal sense and empirically.

**Justification For Why Not Lower Score:**

N/A

---

### Decision · Program_Chairs · 2024-01-16

Reject